# Circulating tumor DNA reveals complex biological features with clinical relevance in metastatic breast cancer

Aleix Prat [1,2,3,4,5,6,16] ✉, Fara Brasó-Maristany [1,16], Olga Martínez-Sáez [1,3,4,16], Esther Sanfeliu[1,7], Youli Xia[2], Meritxell Bellet [6,8,9], Patricia Galván[1], Débora Martínez[1], Tomás Pascual [1,3,6,10], Mercedes Marín-Aguilera[2], Anna Rodríguez[1,3], Nuria Chic[1,3], Barbara Adamo[1,3,4], Laia Paré[2], Maria Vidal [1,3,4], Mireia Margelí[11], Ester Ballana[12], Marina Gómez-Rey[13], Mafalda Oliveira [6,8,9], Eudald Felip[11,12], Judit Matito[13], Rodrigo Sánchez-Bayona[14], Anna Suñol[8,9], Cristina Saura [6,8,9], Eva Ciruelos[14], Pablo Tolosa [14], Montserrat Muñoz[1,3,4,6], Blanca González-Farré[1,7], Patricia Villagrasa[2], Joel S. Parker[15], Charles M. Perou[15] & Ana Vivancos[13]

Liquid biopsy has proven valuable in identifying individual genetic alterations; however, the ability of plasma ctDNA to capture complex tumor phenotypes with clinical value is unknown. To address this question, we have performed 0.5X shallow whole-genome sequencing in plasma from 459 patients with metastatic breast cancer, including 245 patients treated with endocrine therapy and a CDK4/6 inhibitor (ET + CDK4/6i) from 2 independent cohorts. We demonstrate that machine learning multi-gene signatures, obtained from ctDNA, identify complex biological features, including measures of tumor proliferation and estrogen receptor signaling, similar to what is accomplished using direct tumor tissue DNA or RNA profiling. More importantly, 4 DNA-based subtypes, and a ctDNA-based genomic signature tracking retinoblastoma loss-of-heterozygosity, are significantly associated with poor response and survival outcome following ET + CDK4/6i, independently of plasma tumor fraction. Our approach opens opportunities for the discovery of additional multi-feature genomic predictors coming from ctDNA in breast cancer and other cancer-types.

Sequencing of tumor DNA has brought many new biomarkers and possibilities to precision oncology[1]. Detection of somatic gene mutations, amplifications, and gene fusions allows the delivery of targeted therapies in multiple cancer-types, such as lung cancer, colorectal, melanoma and breast cancer[1]. In addition, detection of a high number of somatic mutations (i.e., tumor mutational burden), or a microsatellite instability-high phenotype, can help identify candidates for anti-PD1/PDL1 immune checkpoint inhibitors[2,3]. Importantly, sequencing of tumor DNA in blood samples (i.e., the so-called liquid biopsy,

and henceforth called "ctDNA") allows an easy access to some tumor-based genetic information at any given timepoint and can replace a tumor tissue biopsy in some cases, thus avoiding delays and complications of a solid tumor invasive biopsy procedure, which can be quite challenging sometimes in the metastatic setting.

Identification of single tumor DNA alterations can be clinically useful[1]. However, cancer is highly complex and additional biological information is likely needed to refine the prediction of patients´ prognosis and/or treatment benefit[4]. Breast cancer is the perfect example

---

since RNA-based profiling tests provide clinical and biological useful information beyond individual somatic gene mutations or amplifications such as *PIK3CA* or *ERBB2*[5–8]. In early disease, multi-gene RNA-based prognostic assays (e.g., OncotypeDX, Mammaprint and Prosigna) are available and recommended by clinical guidelines. In advanced disease, RNA-based profiling is becoming a promising prognostic and predictive tool[5–7,9]. Unfortunately, tissue samples in patients with advanced disease are not readily available and, even so, the type of metastatic organ or site can compromise the expression patterns obtained from bulk RNA and might not reflect the intra-patient tumor heterogeneity.

To help overcome these challenges, we previously reported a supervised learning integrative computational approach to predict RNA-based tumor expression signatures values using data from DNA copy-number alterations[10]. Specifically, we found 150 multi-feature DNA-based signatures tracking a variety of breast cancer biological processes, including proliferation status, clinical ER tumor status, tumor histology, and estrogen-signaling pathway activity, with a high predictive performance (i.e., area under the ROC curve ≥0.75)[10]. This finding opened the possibility to use DNA sequencing of ctDNA in blood to capture clinically relevant information beyond single genetic alterations and tumor fraction, the latter being associated with patient's prognosis[11]. Our approach could be highly relevant in the metastatic setting, where ctDNA might be the only readily available genetic material from tumors.

Here, we hypothesized that (1) DNA-based signatures tracking breast cancer biological processes can be detected in ctDNA and provide clinically useful information and (2) DNA-signatures detected in plasma and tissue can help identify tumor subtypes within hormone receptor-positive and HER2-negative (HR+/HER2-) breast cancer. Our work demonstrates that complex tumor phenotypic traits can be identified in ctDNA and might be clinically relevant in advanced breast cancer. Our ctDNA-based multi-gene signature approach opens potential avenues for discovering and implementing biomarkers in oncology, especially in the advanced disease setting where liquid biopsies are becoming commonplace.

## Results

To demonstrate that ctDNA can capture complex tumor phenotypes, 150 previously defined multi-feature DNA-based signatures (Table 1) were applied[10] on 0.5X shallow whole genome sequencing (shWGS) data from a first dataset, hereafter Plasma-1 cohort (Fig. 1a and Supplementary Fig. 1a), composed of 246 plasma samples. Most samples profiled ($n = 209$) were from 174 patients with advanced HR+/HER2- breast cancer. Samples from the same patient were obtained at different timepoints. Additional samples from the other clinical subtypes were also assayed including 19 plasma samples from 16 patients with advanced HER2-positive (HER2+) disease, 17 plasma samples from 16 patients with advanced triple-negative breast cancer (TNBC) and 1 plasma sample from 1 patient with advanced disease but unknown HR and HER2 status.

### Plasma tumor fraction

Of 246 plasma samples, 178 (72.4%) had a Tumor cell Fraction (TF) of ≥3% (range 4–84%; median 9.4%), indicating presence of tumor, according to the ichorCNA tool[12] (Supplementary Fig. 1b). The TFs detected in our study are in line with those reported in other breast cancer studies[13,14]. In plasma samples with a TF ≥3%, we measured the signals of 514 DNA segments and subsequently determined the scores for each of the 150 previously reported DNA signatures designed to predict tumor RNA- and protein-based phenotypes[10] (Fig. 2a). Of note, all signatures/models were applied exactly as previously reported[10] (Supplementary Data 1 and 2); thus, in this study, no new models were developed and the samples analyzed can be considered 'test/validation' datasets.

As expected, TF as a continuous variable was found strongly correlated with the number of altered DNA copy-number segments in each sample (Pearson's rho = 0.76) (Fig. S2). In addition, TF was found strongly correlated (i.e., Pearson's rho ≥0.70 or ≤−0.70) with 46 of 150 (31.0%) ctDNA-based signature scores (Supplementary Data 3). The ctDNA-based signatures highly correlated with TF were mostly tracking biological processes associated with Luminal B disease (i.e., positively correlated with TF) or Luminal A disease (i.e., negatively correlated with TF). To compare the ctDNA-based signature scores across patients, we used the TF-adjusted tumor copy-number signal, as provided by the ichorCNA tool[12]. As expected, this normalization step decreased the strength of association between TF and the number of altered copy-number segments, and between TF and each ctDNA-based signature

## Table 1 | Description of selected DNA-based signatures

| Description | DNA-based signatures | | Original RNA-based signatures | | | |
| --- | --- | --- | --- | --- | --- | --- |
| | Name | DNA segments | N genes | PMID | References | |
| Research-based Mammaprint | NKI70 | 149 | 60 | 11823860 | 46 | |
| Estrogen-regulated gene expression | Scorr_IE | 166 | 754 | 16505416 | 47 | |
| TP53 mutational status | P53_Mut_Correlation | 240 | 48 | 17150101 | 48 | |
| Genes up-regulated in Basal-like | GSEA_SMID_Basal_UP | 232 | 648 | 18451135 | 49 | |
| Proliferation/cell cycle | Wirapati_Proliferation | 112 | 355 | 18662380 | 50 | |
| Retinoblastoma loss of heterozygosity | RB-LOH | 236 | 345 | 18782450 | 16 | |
| Research-based PAM50 Luminal A | Scorr_LumA_Correlation | 197 | 50 | 19204204 | 15 | |
| Research-based PAM50 Luminal B | Scorr_LumB_Correlation | 184 | 50 | 19204204 | 15 | |
| Research-based PAM50 HER2-enriched | Scorr_Her2_Correlation | 66 | 50 | 19204204 | 15 | |
| Research-based PAM50 Basal-like | Scorr_Basal_Correlation | 229 | 50 | 19204204 | 15 | |
| Glycolysis gene expression | Glycolysis_Signature | 123 | 5 | 19291283 | 51 | |
| Luminal-related gene expression | Luminal_Cluster | 236 | 78 | 21214954 | 18 | |
| HER2 amplicon gene expression | HER2_Amplicon | 34 | 19 | 21214954 | 18 | |
| TP53 mutational status in ER + | P53_ER + | 89 | 29 | 21248301 | 52 | |
| P53null luminal mouse model | MM_p53null_Luminal | 166 | 224 | 24220145 | 20 | |
| MYC amplified mouse model | MM_Myc | 184 | 228 | 24220145 | 20 | |
| Estrogen receptor signaling | GP7_Estrogen_signaling | 255 | 701 | 25109877 | 19 | |

The information for each of the 150 ctDNA-based signatures, including the weights of each DNA segment, can be found in Supplementary Data 11.

**a**

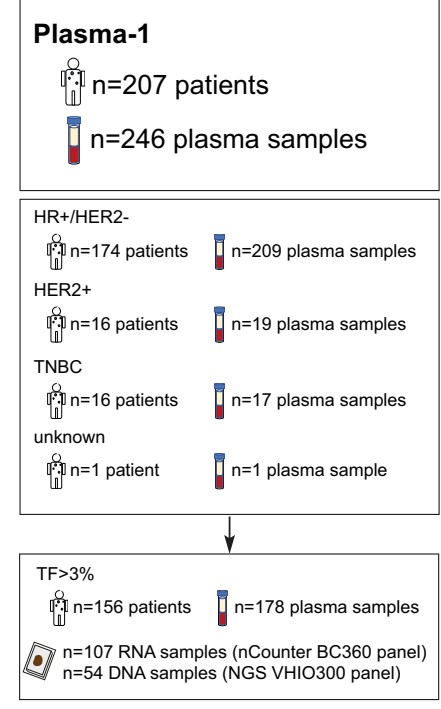

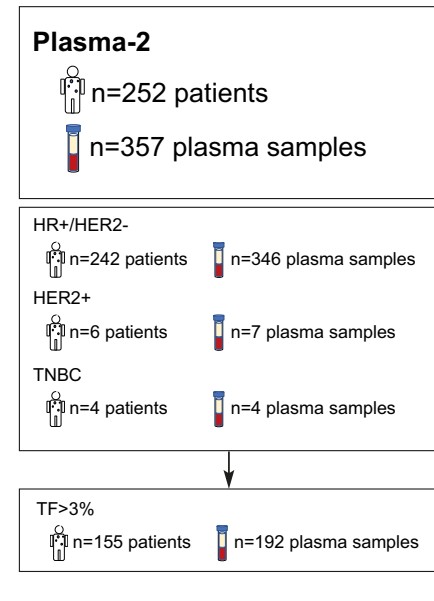

**b**

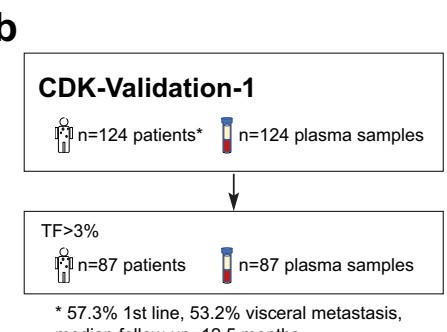

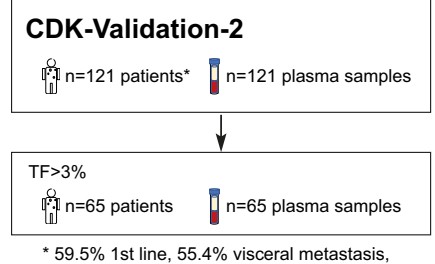

**Fig. 1 | Summary of plasma samples used in the study. a** Description of plasma samples obtained from patients with advanced breast cancer across Plasma-1 and Plasma-2 cohorts. **b** Description of samples from patients with advanced HR+/HER2- breast cancer treated with endocrine therapy and a CDK4/6 inhibitor (CDK-Validation-1 and CDK-Validation-1 cohorts). TF tumor fraction.

score (Supplementary Data 3). Of note, no ctDNA-based signature was found moderately or strongly correlated with TF (i.e., Pearson´s rho ≥0.50) after the adjustment (Supplementary Data 3).

**Plasma versus tissue DNA-based signatures**

We next explored the correlation of each of the 150 DNA-based signatures determined using plasma ctDNA (adjusted by TF) versus tumor tissue DNA across 54 patients with available paired sample-types obtained within a timeframe of 8-weeks ($n = 27$) or more than 8 weeks ($n = 27$). Genome-wide copy number data was obtained from formalin-fixed paraffin-embedded (FFPE) tumors using a capture-based panel, and from ctDNA using shWGS (see methods). Across all 150 signatures, the average correlation coefficient between tumor and plasma was 0.40 (range 0.02 to 0.66) and 40 signatures (26.7%) had a correlation coefficient ≥0.50. When the correlations were evaluated in the 27 cases where plasma and tumor were obtained within a timeframe of 8.0 weeks, the number of signatures with a correlation coefficient ≥0.50 was 63 (42% versus 19.3% in the 27 cases where plasma and tumor were obtained in >8.0 weeks; $p < 0.001$), including 36 of the 40 (90%) signatures with a correlation coefficient ≥0.50 in all patients

(Fig. 2b and Supplementary Data 4). Two of the highly correlated DNA signatures were the UNC_8q_Amplicon and the UNC_Scorr_P53_Mutation. Overall, these results suggest a moderate association between ctDNA-based and tumor DNA-based signatures across timepoints and DNA sequencing approaches (i.e., ctDNA shWGS versus capture-based using FFPE DNAs). Of note, immune-related DNA-based signatures showed low correlation coefficients (Supplementary Data 4).

To further demonstrate that copy number alterations (CNA)-based data from plasma allows the identification of similar biological states as in tissue, we calculated the intra-patient correlation of the CNA-based signals of 514 DNA segments across 54 paired samples (tumor tissue versus plasma). Overall, 57% of patients had a correlation coefficient between plasma and tissue >0.50, which increased to 83% when we evaluated 29 patients with plasma TF >10%. In these 29 patients with a plasma TF >10%, 59 and 24% had a correlation coefficient of >0.70 and >0.80, respectively (Fig. 2c and Supplementary Data 5). Overall, these results strongly suggest that plasma ctDNA can reliably capture CNA-based signals from tumor tissue, although the amount of ctDNA might impact the ability to accomplish this (Fig. 2d and Fig. S3).

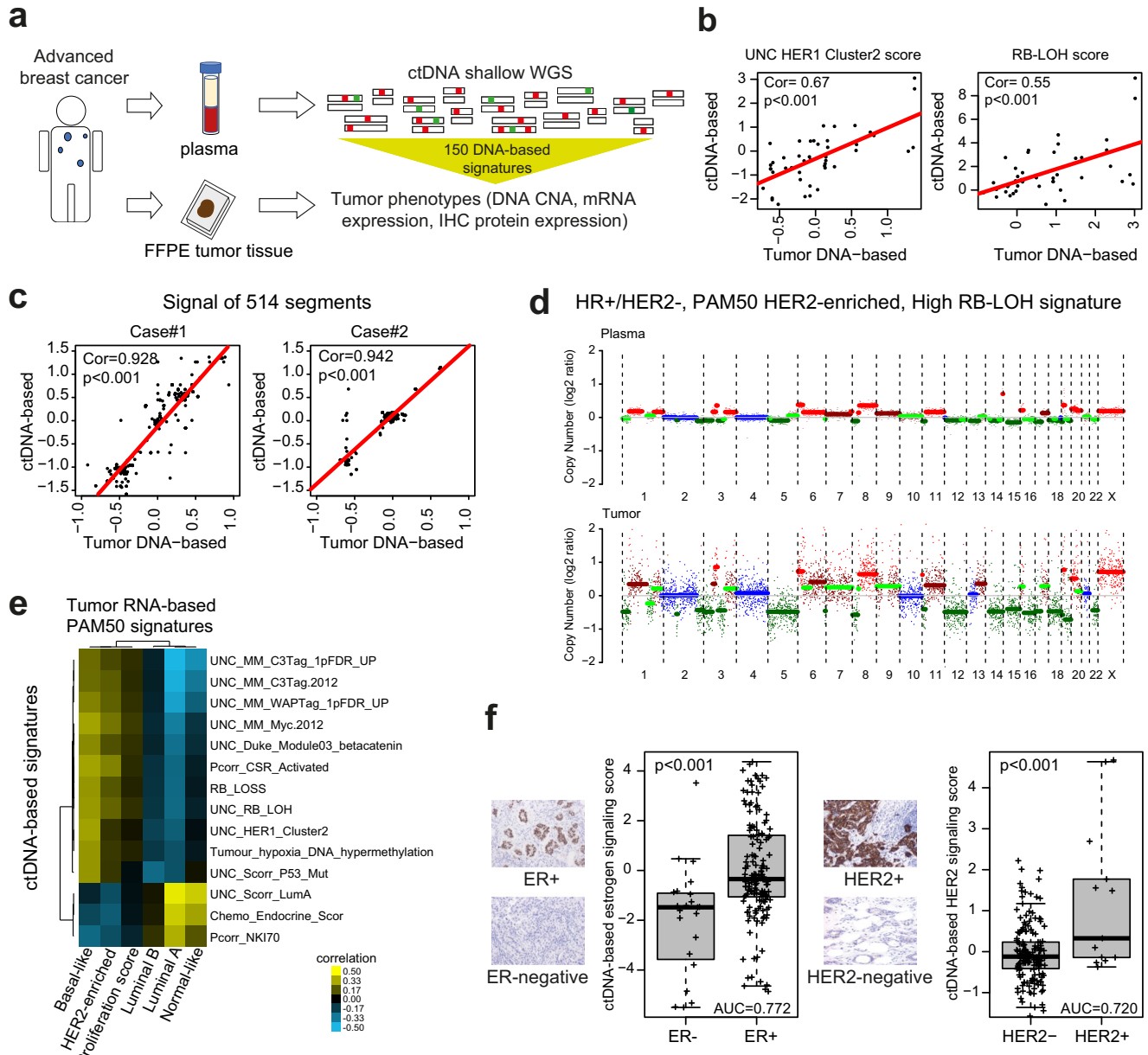

**Fig. 2 | Circulating tumor DNA (ctDNA) in metastatic breast cancer. a** Plasma samples were obtained from 207 patients (174 with HR+/HER2-, 16 HER2+, 16 TNBC, 1 N/A). After purification of plasma cell-free DNA, shallow whole genome sequencing (shWGS) was performed. Using the ctDNA-based sequencing data from 514 DNA segments, 150 previously developed DNA copy number-based signatures[10] tracking a variety of biological processes were applied in patients with a Tumor cell Fraction (TF) ≥3%. Individual scores for each signature were obtained. The complete names of the signatures, references and gene lists can be found in Supplementary Data 1. **b** Examples of correlation between the scores of the UNC_HER1_Cluster2 ($p < 0.001$) and the RB-LOH ($p < 0.001$) signatures when determined in plasma versus tumor tissue (the correlations for the other signatures can be found in Supplementary Data 4). Of note, tissue samples were obtained at different timepoints than the plasma samples. **c** Examples of correlations of 514 DNA signals between plasma and tissue in two single patients (case#1 $p < 0.001$, case#2 $p < 0.001$). Correlation coefficients (Cor) and $p$ in **b** and **c** were determined by Pearson correlation. **d** Example of copy number alterations (CNA) plots of matched timepoint plasma and tissue samples of a HR+/HER2- tumour, PAM50 HER2-enriched and high RB-LOH signature score. **e** Unsupervised cluster heatmap analysis of Pearson´s correlation coefficients obtained by comparing the scores of the top individual ctDNA-based signature versus the scores of each individual PAM50 RNA-based tissue signature across 58 matched-timepoint paired plasma-tissue cases. Correlation values can be found in Supplementary Data 6. **f** Boxplots of a ctDNA-based signatures tracking ER-related biology in ER-negative ($n = 23$) and ER+ tumors ($n = 155$) ($p < 0.001$) (left) and boxplots of a ctDNA-based signatures tracking HER2-related biology in HER2- ($n = 165$) and HER2+ tumors ($n = 13$) ($p < 0.001$) (right). $P$-value ($P$) was determined by two-tailed unpaired t-tests. For the boxplot, center line indicates median; box limits indicate upper and lower quartiles; whiskers indicate 1.5× interquartile range. Examples of ER and HER2 immunohistochemistry (ICH) stainings (20X) are provided. Source data are provided as a Source Data file.

## ctDNA-based data versus tissue RNA-based expression data

RNA-based expression data from FFPE tissue using a research-based PAM50 intrinsic subtype assay[15] was available for 108 cases with a TF ≥3% in plasma. Tissue samples were obtained at various timepoints. To further explore the association between ctDNA enrichments and RNA expression data, we evaluated the correlation of 6 PAM50 RNA-based tissue signatures with each of the 150 ctDNA-based signatures (adjusted by TF) (Supplementary Data 6). To summarize these results, we plotted in an unsupervised cluster analysis, the correlation coefficients of the most correlated

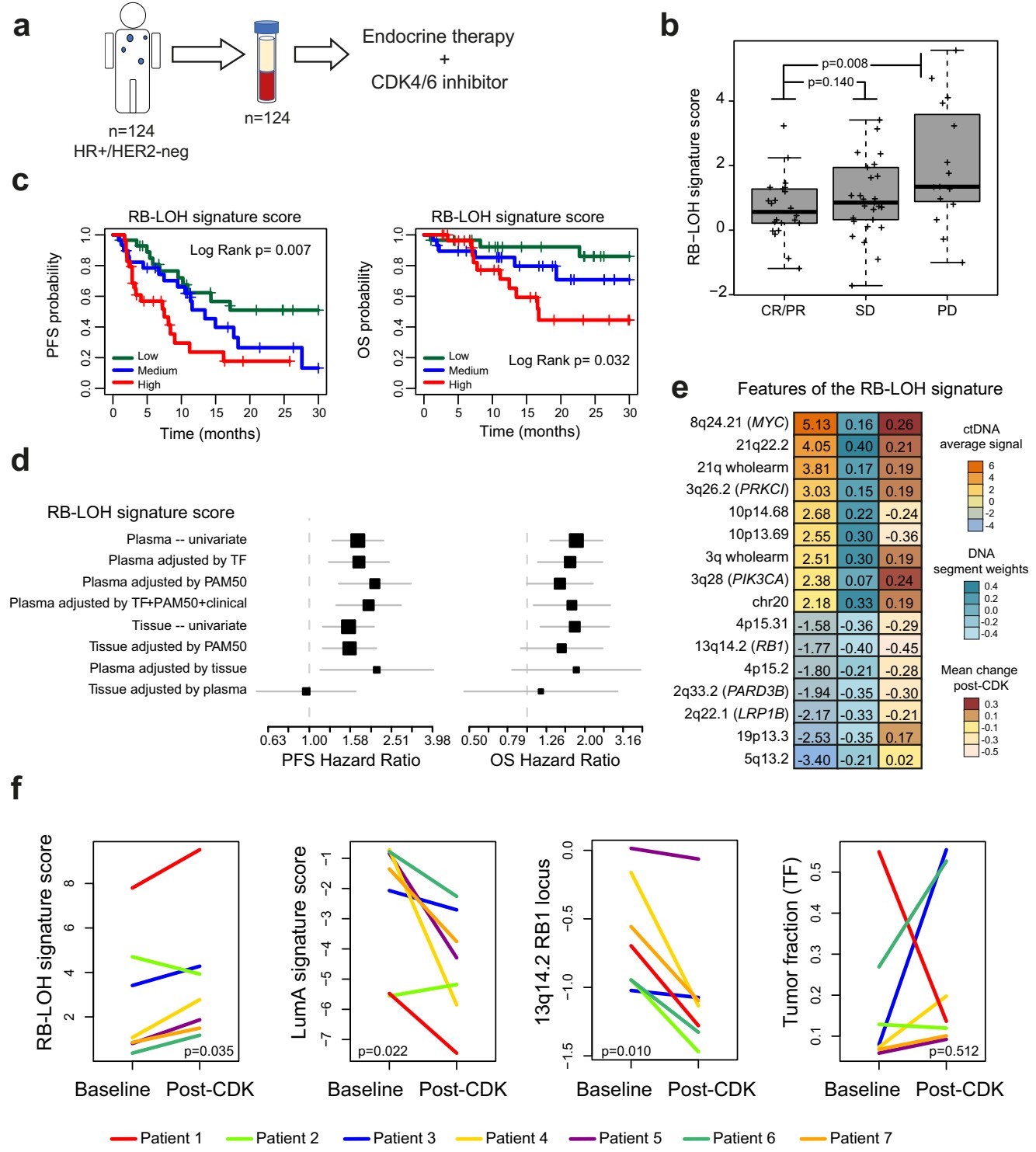

signatures across 58 matched-timepoint paired plasma-tissue cases (Fig. 3c). In general, ctDNA-based signatures were positively and negatively correlated with the known biology that each PAM50 subtype signature is hypothesized to be tracking. For example, a ctDNA-based signature tracking RB-LOH gene expression signature[16], which is enriched in E2F target genes and tracks tumor proliferation rates (Supplementary Fig. 4), was found positively correlated to the PAM50 RNA-based Basal-like, HER2-enriched and proliferation tumor signatures, and negatively correlated to the PAM50 RNA-based Luminal A tumor signature (Fig. 2e).

Beyond PAM50, we evaluated the expression of 771 genes in tumors using the nCounter Breast Cancer 360 Panel in 107 cases

with a TF ≥3% in plasma. Correlation coefficients of the mRNA expression of each individual gene with each 150 ctDNA-based signature scores were also determined. Like the PAM50 RNA tumor signatures, the mRNA expression of luminal genes (e.g., *ESR1* and *GATA3*) was positively correlated with luminal ctDNA-based signatures, while proliferation and cell cycle-related genes by mRNA (e.g., *MKI67, AURKA, TTK, E2F1 and CCNE1*) were positively correlated with proliferation-related ctDNA-based signatures (Supplementary Fig. 5). Overall, these findings confirm that DNA-copy number-based signatures coming from ctDNA can track the main breast cancer phenotypes and their known gene expression features.

**Fig. 3 | The ctDNA-based RB-LOH signature predicts clinical outcome in advanced HR+/HER2- breast cancer treated with endocrine therapy and a CDK4/6 inhibitor. a** A plasma sample was obtained from 124 patients within 48 h prior to starting endocrine therapy and CDK4/6 inhibition (CDK-Validation-1 cohort). ctDNA-based signatures were applied in plasma samples with a TF ≥3% (*n* = 87). **b** Boxplots of RB-LOH ctDNA-based signature score in patients with complete or partial response (CR/PR, *n* = 26), stable disease (SD, *n* = 32) and progressive disease (PD, *n* = 18). For the boxplot, center line indicates median; box limits indicate upper and lower quartiles; whiskers indicate 1.5× interquartile range. *P*-values (*p*) were determined by two-tailed unpaired *t*-tests. **c** Kaplan-Meier curves of PFS (left) and OS (right) of the RB-LOH ctDNA-based signature. Each patient group is based on tertiles. *P*-values (*p*) were determined by Log Rank Test. **d** Forest plots of hazard ratios (HRs) for PFS (left) and OS (right) of the RB-LOH DNA-based signature when evaluated in plasma alone (i.e., plasma–univariate; *n* = 87), in

plasma when adjusted for TF (*n* = 87), in plasma when adjusted for PAM50 RNA-based subtypes (*n* = 53), in plasma when adjusted for TF + PAM50 + clinical variables (*n* = 53), in tissue alone (i.e., tissue–univariate; *n* = 63) and in plasma when adjusted for tissue and vice-versa (*n* = 28). Data are presented as the hazard ratios (HR) with error bars showing 95% confidence intervals. **e** Average ctDNA signal of 16 features of the RB-LOH DNA-based signature (column on the left), weight and direction of each feature (column in the middle) in the original signature as reported in Xia et al.[10] and mean change of the 16 features (column on the right) in 7 patients with paired plasma samples (baseline pre-treatment vs post-CDK4/6 inhibitor treatment). **f** ctDNA-based signature scores of the RB-LOH signature, the Luminal A signature, the 13q14.2 *RB1* locus, and TF across 7 patients with paired plasma samples (baseline vs post-CDK4/6 inhibitor treatment). *P*-values (*p*) were determined by two-tailed paired t-tests. Source data are provided as a Source Data file.

## ctDNA-based signatures versus tissue ER and HER2 status

ER expression by immunohistochemistry (IHC), and HER2 over-expression by IHC and/or amplification by in-situ hybridization, are key biological features of breast cancer[4,17]. To evaluate the relationship between ctDNA-based information and ER or HER2 tumor clinical biomarker status, we evaluated the association of each of the 150 adjusted ctDNA-based signatures with either ER clinical status (i.e., positive versus negative) or HER2 status (i.e., positive versus negative) across 177 of 178 samples with a TF ≥3% with known ER and HER2 IHC status from tumor tissue. As expected, ctDNA-based signatures tracking luminal biological processes (e.g., luminal_cluster_signature[18] and GSEA_Median_GP7_Estrogen_signaling[19]) were found enriched in ER-positive (ER+) disease (*p* < 0.001; false discovery rate [FDR] < 1%; highest AUC = 0.77) compared to ER-negative disease. Similarly, ctDNA-based signatures tracking HER2 expression or amplification (e.g., HER2-signature[15] and HER2-amplified-HER2-amplicon[18]) were found significantly enriched (*p* < 0.001; FDR < 1%; highest AUC = 0.72) in HER2+ disease compared to HER2- disease (Fig. 2f and Supplementary Fig. 6). Overall, these results suggest that ctDNA-based profiling captures and predicts specific phenotypic tumor traits.

## ctDNA-based signatures in metastatic HR+/HER2- disease treated with endocrine therapy and a CDK4/6 inhibitor

To evaluate the association of each ctDNA-based signatures with patient's outcome, we analyzed baseline pre-treatment plasma samples from 124 patients with advanced HR+/HER2- breast cancer treated with endocrine therapy and a CDK4/6 inhibitor (CDK-Validation-1 cohort). Eighty-seven plasma samples had a TF ≥3% (Fig. 1b and Fig. 3a). The median follow-up was 12.5 months (range 1.0 to 56.7 months), and most patients were defined as endocrine-sensitive (83.9%) and treated in the first-line setting (59.8%) (Table 2). TF was associated with poorer progression-free survival (PFS) but was not associated with overall survival (OS) (Supplementary Fig. 7). Among the different clinical variables, treatment line (i.e., first versus second/third) and bone-only disease were associated with better PFS, while ECOG performance status was the only variable associated with OS (Supplementary Fig. 8).

From the 150 ctDNA-based signatures (adjusted by TF), 36 (24%) and 37 (25%) were found significantly associated with PFS and OS, respectively (Supplementary Data 7). Twenty-seven (18%) signatures were found significantly associated with both PFS and OS. In general, signatures associated with poor survival outcome were those hypothesized to be tracking proliferation and non-ER+/non-luminal-related biological processes, such as the MM_p53null.-Luminal (i.e., TP53-deficient) and MM_Myc signatures (i.e., high *c-MYC* amplification)[20]. Conversely, ctDNA-signatures associated with better outcome were tracking luminal A-related biological processes. Similar results were obtained when the analysis was performed in the subset of 52 patients treated with endocrine

therapy and a CDK4/6 inhibitor in the first-line setting (Supplementary Data 7 and Supplementary Fig. 5).

A high score of a ctDNA-signature tracking RB-LOH(19) was found one of the top biomarkers associated with poor outcome and treatment response (Fig. 3b, c). Since loss of RB is a known mechanism of resistance to CDK4/6 inhibitors[21–25], we decided to focus on this signature, which is composed of 224 copy number features, including amplification of 2p (e.g., *ETV6*), 3q (e.g., *PIK3CA*), 8q (e.g., *MYC*), 20q (e.g., *AURKA*) and 21q (e.g., *TMPRSS2* and *ERG*), and deletion of 2q (e.g., *PARD3B*), 4q, 5q, 12q, 13q (e.g., *RB1*), 15q and 17p. Interestingly, multivariable analysis showed that the association of the ctDNA RB-LOH signature with PFS and OS was independent of TF (as a continuous variable), type of CDK4/6 inhibitor, line of treatment (first-line versus second-line versus later lines), presence of visceral disease and number of metastatic sites (Fig. 3d). Finally, the direction (i.e., amplification or deletion) and strength (i.e., coefficient) of the 48 main features of the original tissue-based DNA RB-LOH signature[10] were properly detected in ctDNA (correlation coefficient between the original coefficient weights of the 48 main DNA segments of the RB-LOH signature and the actual ctDNA signals measured in plasma = 0.72, *p* < 0.001; Fig. 3e and Supplementary Fig. 10).

## RB-LOH in tumor tissue versus plasma in metastatic HR+/HER2- disease treated with endocrine therapy and a CDK4/6 inhibitor

We analyzed tissue samples of 63 patients with advanced HR+/HER2- breast cancer treated with endocrine therapy and a CDK4/6 inhibitor. RB-LOH signature determined in tumor tissue DNA was significantly associated with PFS and OS (Fig. 3d and Supplementary Fig. 11). Twenty-eight patients had paired tumor tissue and plasma samples, these were interrogated to compare the ability of the tissue-based signatures versus the ctDNA-based signatures to better predict PFS and OS. Of the 150 DNA-based signatures, 17% and 13% were statistically significantly associated with PFS and OS, respectively, when evaluated in plasma. When the same signatures were evaluated in tissue, only 2% and 1% were statistically significantly associated with PFS and OS, respectively (Supplementary Data 8). Thus, ctDNA-based signatures were better in predicting survival outcomes in advanced breast cancer treated with endocrine therapy and CDK4/6 inhibition than the same signatures when evaluated in tissue, suggesting that plasma-based signatures capture "the most up to date" biological state of the disease before starting therapy. Furthermore, when the RB-LOH signature was evaluated head-to-head in the 28 patients with paired tumor tissue and plasma samples in a bivariate cox model, the RB-LOH ctDNA plasma signature was found significantly associated with PFS, but not the RB-LOH tumor signature (Fig. 3d). Overall, baseline pre-treatment ctDNA better captured the prognosis of patients than archival tumor tissue DNA.

**Table 2 | Baseline clinical characteristics of patients with HR+/HER2- advanced disease treated with endocrine therapy and a CDK4/6 inhibitor**

| Cohort | CDK-Validation-1 | | | | CDK-Validation-2 | | | |
|---|---|---|---|---|---|---|---|---|
| | All patients | | Patients with TF ≥3% | | All patients | | Patients with TF ≥3% | |
| | *N* | % | *N* | % | *N* | % | *N* | % |
| Number of patients | 124 | | 87 | | 121 | | 65 | |
| Median age (range) | 61 (34-86) | | 62 (34-86) | | 60 (31-88) | | 60 (41-83) | |
| Sex | | | | | | | | |
| Female | 122 | 98.4% | 87 | 100.0% | 121 | 100% | 121 | 100% |
| Male | 2 | 1.6% | 0 | 0.0% | 0 | 0% | 0 | 0% |
| CDK4/6 inhibitor | | | | | | | | |
| Palbociclib | 70 | 56.5% | 48 | 55.2% | 48 | 39.7% | 22 | 33.8% |
| Ribociclib | 50 | 40.3% | 36 | 41.4% | 52 | 43.0% | 26 | 40.0% |
| Abemaciclib | 4 | 3.2% | 3 | 3.5% | 21 | 17.4% | 17 | 26.2% |
| Setting in advanced disease | | | | | | | | |
| 1st line | 71 | 57.3% | 52 | 59.8% | 72 | 59.5% | 37 | 56.9% |
| 2nd line | 27 | 21.8% | 15 | 17.2% | 18 | 14.9% | 10 | 15.4% |
| ≥3rd line | 26 | 21.0% | 20 | 23.0% | 31 | 25.6% | 18 | 27.7% |
| Number of metastases | | | | | | | | |
| <3 | 74 | 59.7% | 50 | 57.5% | 56 | 46.3% | 44 | 67.7% |
| ≥3 | 50 | 40.3% | 37 | 42.5% | 64 | 52.9% | 21 | 32.3% |
| Unknown | 0 | 0.0% | 0 | 0.0% | 1 | 0.8% | 0 | 0.0% |
| Type of metastasis | | | | | | | | |
| Visceral metastasis | 66 | 53.2% | 34 | 39.1% | 67 | 55.4% | 38 | 58.5% |
| De novo metastasis | 30 | 24.2% | 19 | 21.8% | 34 | 28.1% | 22 | 33.9% |
| Bone only | 12 | 9.7% | 0 | 0.0% | 20 | 16.5% | 12 | 18.5% |
| Prior endocrine therapy sensitivity | | | | | | | | |
| Sensitive | 103 | 83.1% | 73 | 83.9% | 94 | 77.7% | 48 | 73.8% |
| Resistant | 19 | 15.3% | 14 | 16.1% | 26 | 21.5% | 17 | 26.2% |
| Unknown | 2 | 1.6% | 0 | 0.00% | 1 | 0.8% | 0 | 0.0% |
| ECOG Performance status | | | | | | | | |
| 0 | 50 | 40.3% | 38 | 43.7% | 62 | 51.2% | 33 | 50.8% |
| 1 | 61 | 49.2% | 39 | 44.8% | 49 | 40.0% | 27 | 41.5% |
| 2 | 12 | 9.7% | 10 | 11.5% | 5 | 4.1% | 4 | 6.2% |
| Unknown | 1 | 0.8% | 0 | 0.0% | 5 | 4.1% | 1 | 1.5% |

*Previous systemic treatments are available for the CDK-Validation-1 cohort (Supplementary Data 17).

### ctDNA RB-LOH signature versus ctDNA RB1 individual region in metastatic HR+/HER2- disease treated with endocrine therapy and a CDK4/6 inhibitor

The DNA-based RB-LOH signature considers the signal of the *RB1* locus (13q14.2) among 224 other features[10] (Fig. 3e). The correlation coefficient between the ctDNA signal of 13q14.2 and the ctDNA RB-LOH signature score was −0.12 across the 178 samples with TF ≥3%, reinforcing the concept that the ctDNA RB-LOH signature is different from measuring only the individual 13q14.2 region. Indeed, the ctDNA signal of the individual 13q14.2 segment was not significantly associated with PFS ($p = 0.061$) but was significantly associated with better OS ($p = 0.020$), while the RB-LOH signature was strongly associated with PFS and OS in patients with advanced HR+/HER2-negative breast cancer treated with CDK4/6 inhibitors. In a bivariate cox model with both variables, only the ctDNA RB-LOH signature was prognostic. Overall, the RB-LOH ctDNA-based signature better captured the clinical behavior than an individual DNA region looking only at *RB1*, thus highlighting the power of a multi-feature algorithm for sensing pathway activity (Supplementary Fig. 12).

### Independent prognostic validation of the RB-LOH signature

To further validate the prognostic value of the RB-LOH signature, we evaluated pre-treatment baseline plasma samples from a second independent cohort of 357 plasma samples, hereafter Plasma-2 cohort, which included 121 patients with advanced HR+/HER2- breast cancer treated with endocrine therapy and a CDK4/6 inhibitor at Hospital Clínic of Barcelona and the Vall d'Hebron Institute of Oncology (VHIO) and assayed using our ctDNA WGS-assay. Baseline characteristics of the 121 patients of CDK-Validation-2 cohort are reported in Table 2. Sixty-five patients (54.0%) had a TF ≥3%. With a median follow-up of 12.3 months, the RB-LOH signature was significantly associated with worse PFS when evaluated as a continuous variable (hazard ratio=1.42 [1.03–1.96], $p = 0.032$), and using the previously defined cutoffs (i.e., tertiles) derived from the CDK-Validation-1 cohort (Supplementary Fig. 13a, b). Finally, the prognostic value of RB-LOH signature was confirmed in the combined CDK-Validation-1 and CDK-Validation-2 cohorts ($n = 152$) (Supplementary Fig. 13c, d). Moreover, we evaluated if the ctDNA-based RB-LOH was significantly associated with PFS and OS in 71 patients with a TF 3–10% before starting endocrine therapy in combination with a CDK4/6 inhibitor. Like the overall population, we observed a statistically significant association with both clinical endpoints (PFS: hazard ratio = 1.32, $p = 0.023$; OS: Hazard ratio = 1.54, $p = 0.011$), suggesting that our approach can work in patients with low TF (3–10%).

To provide more evidence of the prognostic value of the RB-LOH signature, we evaluated publicly available data from an independent cohort of 381 patients with advanced HR+/HER2- breast cancer treated with endocrine therapy and a CDK4/6 inhibitor at the Memorial Sloan Kettering Cancer Center (MSK)[26]. RB-LOH signature scores were obtained from next-generation-sequencing data of tumor tissue DNA profiled under the MSK-IMPACT platform, which includes copy-number data from 468 cancer-associated genes[27]. With a median follow-up of 71.0 months, the RB-LOH signature was significantly associated with PFS (hazard ratio = 1.41, 95% CI 1.21–1.64, $p < 0.001$) (Supplementary Fig. 14a, b). Similar results were obtained in the subset of 223 patients where the MSK-IMPACT assay was performed in biopsies taken within 1 year of starting therapy (PFS hazard ratio = 1.61, 95% CI 1.32–1.96, $p < 0.001$) (Supplementary Fig. 14c). However, RB-LOH was not prognostic when assessed in 158 biopsies taken more than 1 year before starting therapy with a CDK4/6 inhibitor (Supplementary Fig. 14d).

### Capturing biological features before and after CDK4/6 inhibition

Scores from the 150 ctDNA-based signatures (adjusted by TF) were evaluated in paired plasma samples (i.e., baseline versus post-treatment after progressive disease) in 7 patients with advanced HR+/HER2- breast cancer treated with endocrine therapy and a CDK4/6 inhibitor (Fig. 3f). Among them, 103 signatures (68.7%) were found differentially enriched between the two time-points (FDR < 5%). As might be expected, enrichment of signatures tracking non-luminal/proliferation-related biological processes (e.g., RB-LOH) and luminal A-related biological processes were found significantly increased and decreased, respectively, in post-treatment samples compared to pre-treatment samples (Fig. 3f and Supplementary Data 9). Of note, TF did not significantly change between the two timepoints across the 7 patients (Fig. 3f), and 1 patient with a substantial decrease in TF still showed an increase in the RB-LOH score and a decrease of the luminal A signature. Importantly, the signal from 12 of the top 16 DNA segments (75%) of the RB-LOH signature changed in the expected direction in post-treatment samples compared to pre-treatment samples (Fig. 3f), providing evidence of our approach providing additional information beyond changes of TF and signals from individual DNA regions. These biological changes identified in ctDNA are concordant with similar biological changes identified across 18 patients with paired tumor-based RNA expression before and at progression to endocrine therapy and a CDK4/6 inhibitor (Supplemental Fig. S15). Specifically, PAM50 Luminal A and proliferation signatures were found significantly decreased and increased, respectively, in tumor tissue progressing to endocrine therapy and CDK4/6 inhibition.

### Identifying ctDNA-based tumor subtypes

Phenotype-based classification in metastatic breast cancer using tumor tissue RNA expression profiling such as intrinsic subtyping (i.e., Luminal A, Luminal B, HER2-enriched and Basal-like) is prognostic and might predict treatment benefit[5–7]. To evaluate if the biology displayed by the 150 ctDNA-based signatures (adjusted by TF) can identify subtypes with clinical relevance, we performed an unsupervised hierarchical cluster analysis of all 150 signatures across 178 plasma samples with a TF ≥3% (Fig. 4a). Four main clusters/groups of samples were identified using consensus clustering plus (Supplementary Fig. 16) and then validated in an independent cohort of 357 plasma samples (Plasma-2 cohort), including 193 with a TF ≥3% (Supplementary Fig. 17).

Clusters 3 and 4 showed high scores of ctDNA-based proliferation-related signatures and low scores of differentiation status and of luminal A-related signatures. Compared to Cluster 4, Cluster 3 showed high expression of basal-like-related gene expression ($p < 0.001$). Cluster 2 showed high enrichment of differentiation and luminal B-related signatures, and low enrichment of basal-like related biology.

Visually, cluster 2 could be further subdivided (minimum 20 samples and a correlation coefficient >0.75) into Cluster2A and Cluster2B, both of which showed differences in the enrichment of ctDNA-based proliferation features, and luminal A-related signatures. Consistent with the luminal A-related biology identified in Custer 2 A, this group was characterized by 16p amplification and 16q deletion (Supplementary Data 10 and Fig. S18), both of which are known features of low-grade and low-proliferative breast cancers[28–30]. Finally, Cluster 1 showed low enrichment of proliferation and luminal B-related signatures and high enrichment of luminal A subtype related signatures. Plasma TF in Cluster 1 was significantly lower compared to the other clusters combined (average 6.8% vs. 9.8%), $p < 0.001$; Fig. 4a and Supplementary Fig. 19, which might be predicted for slow growing luminal A tumors.

RNA-based expression data from FFPE tissue using a research-based PAM50 intrinsic subtype assay[15] was available for 108 cases with a TF ≥3% in plasma. Tissue samples were obtained at various time-points. As expected, Cluster 3 was enriched for tumors with a PAM50 non-luminal subtype (i.e., HER2-enriched or Basal-like) compared to the other clusters (85.7% versus 25%, $p < 0.001$) (Supplementary Fig. 20). Concordant with this finding, the PAM50 Luminal A and HER2-enriched signatures (as a continuous variable) were found differentially expressed in Cluster 3 versus the other clusters (Fig. 4b). In addition, we observed that Cluster 2B was enriched for PAM50 Luminal B tumors compared to Cluster 2A (53.3% versus 18.8%, $p = 0.044$).

### Prognosis of ctDNA-based tumor subtypes in patients with metastatic HR+/HER2- disease treated with endocrine therapy and a CDK4/6 inhibitor

This work demonstrates that tumor profiles identify samples with similar patterns of expression, and these patterns are associated with clinically relevant genotypes. We then hypothesized that ctDNA-defined subtypes, representing repeatably observed combinations of these patterns, may explain variation in clinical outcomes. Regarding the relationship between clusters 1–4 and treatment response, we observed an overall response rate (ORR) of 52.7%, 34.0%, 7.1% and 16.7% in Cluster 1, Cluster 2, Cluster 3 and Cluster 4, respectively ($p < 0.001$). Regarding the relationship between cluster 1–4 and prior endocrine sensitivity, this was observed in 91.7%, 80.0%, 57.1%, and 55.6% in Cluster 1, Cluster 2, Cluster 3 and Cluster 4, respectively ($p = 0.004$) (Fig. 4c).

Next, we evaluated the prognostic value of the 4 ctDNA-based tumor groups in the CDK-Validation 1 and 2 cohorts combined ($n = 152$). Compared to Cluster 1 (median PFS = 18.3 months), Clusters 2, 3 and 4 were found significantly associated with worse PFS (median PFS of 10.2, 2.9, and 6.4 months, respectively). Median OS was not reached for Cluster 1 and 2, while median OS were 16.8 and 17.8 months for Clusters 3 and 4, respectively. Cluster 3, which has the highest ctDNA-based RB-LOH signature score ($p < 0.001$; Supplemental Fig. S21), was significantly associated with worse PFS and OS compared to Cluster 1 (PFS hazard ratio = 4.76, 2.29–9.91, $p < 0.001$; OS hazard ratio = 5.21, 1.68–16.19, $p < 0.001$) (Fig. 4d).

### DNA-based tumor subtypes in tissue samples

To explore how tumor subtypes identified in ctDNA data perform when applied to tumor tissue DNA, the scores of the 150 DNA-based signatures were determined and evaluated using tumor DNAs from 1,689 patients with early-stage breast cancer from the publicly available METABRIC dataset[29], and 381 patients with advanced HR+/HER2- breast cancer from the MSK-IMPACT dataset (Fig. 5a and Supplemental Fig. 22a). We developed a 4-class subtype classifier from the ctDNA groups identified in Plasma-1 cohort (Fig. 4a) and applied this predictor onto METABRIC's and MSK´s tumor DNA data. Overall, the 4 main clusters were identified in both cohorts (Fig. 5a and Supplemental Fig. 22a), and these clusters showed the expected DNA signal across chromosomes (Fig. 5b). Concordant with this finding, the PAM50 non-

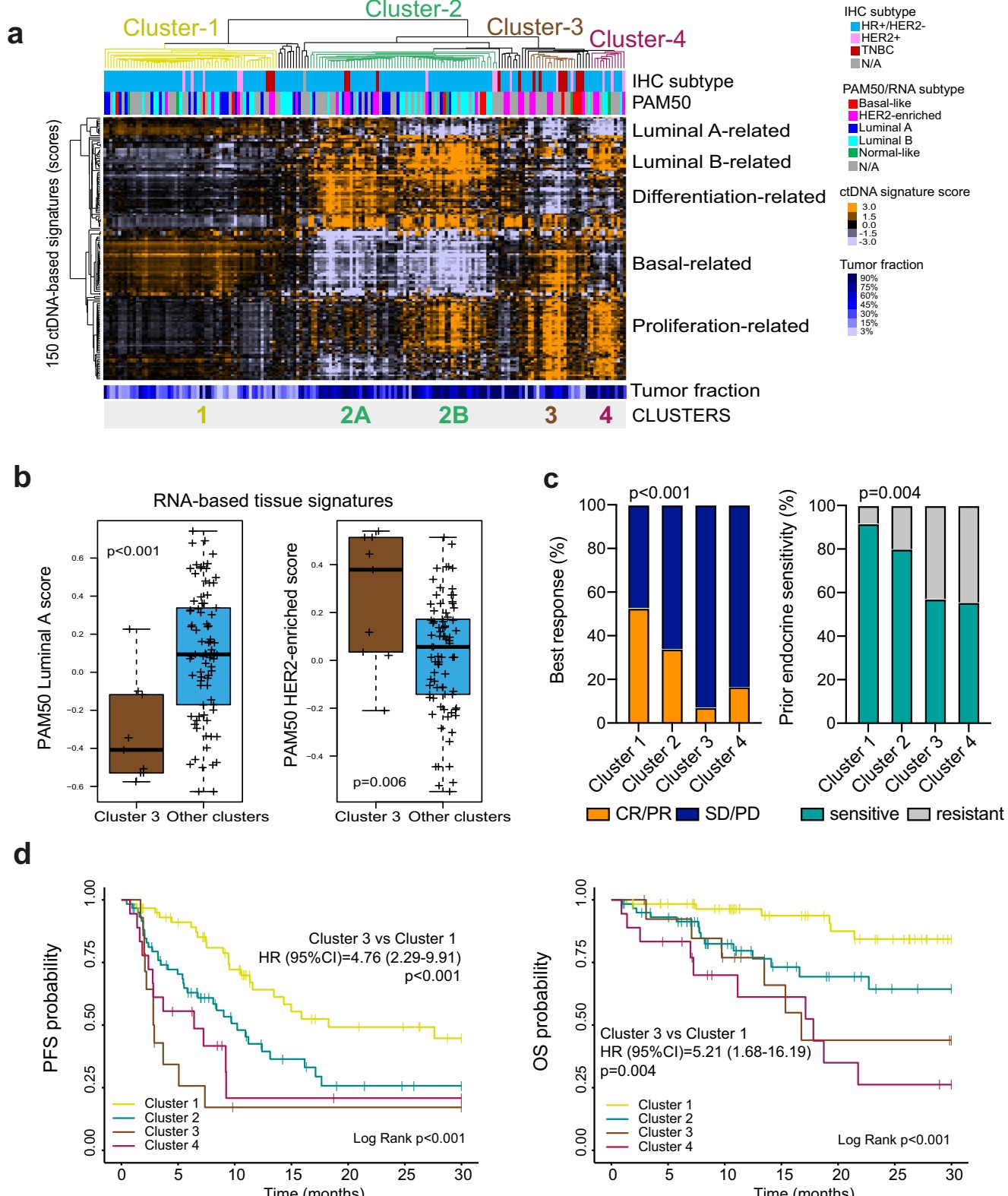

luminal subtypes in the METABRIC cohort were enriched in Cluster 3 compared to the other clusters (79.6% versus 11.6%, *p* < 0.001) (Fig. 5c). In addition, *TP53* somatic mutations in METABRIC were also enriched in Cluster 3 compared to the other clusters (80.7% versus 51.2%, *p* < 0.001) (Fig. 5d). Finally, the 4 main clusters were found significantly associated with disease-free survival (DFS) and OS in all patients, and in patients with early-stage HR+/HER2- breast cancer (Fig. 5e), and PFS in

patients with advanced HR+/HER2- breast cancer in the MSKCC dataset (Supplemental Fig. 22b). Similar results were obtained in the subset of 223 patients where the MSK-IMPACT assay was performed in tumor tissue biopsies obtained within 1 year of initiating therapy (Supplemental Fig. 22c, d).

Finally, to understand whether the identification of the 4 clusters/subtypes is independent of the tissue profiled (i.e., primary versus

**Fig. 4 | ctDNA-based profiling of metastatic breast cancer. a** Unsupervised cluster analysis of 178 plasma samples with a TF ≥3% (columns) and the scores of 150 ctDNA-based signatures (rows). Orange and violet colors represent scores above and below the median score of the signature across the dataset. Below the array tree, the IHC subtype and the PAM50 molecular subtype are shown for each sample. Four clusters of samples (clusters 1 to 4) were identified. Within cluster 2, two subgroups of samples were also identified (clusters 2A and 2B). **b** Expression of 2 tissue PAM50 RNA-based signatures (i.e., Luminal A [$p = 0.0005$] and HER2-enriched [$p = 0.006$]) in cluster 3 versus the other clusters. This analysis was performed in 107 paired plasma and tumor tissue samples. Of note, 58 tumor tissue samples were obtained at the same timepoint as the plasma sample and 49 tumor tissue samples were obtained at different timepoints prior to obtaining the plasma sample. *P*-values (*p*) were determined by two-tailed unpaired *t*-tests. For the box-plot, center line indicates median; box limits indicate upper and lower quartiles; whiskers indicate 1.5× interquartile range. **c** Association of ctDNA-based clusters with response to endocrine therapy and CDK4/6 inhibition ($p < 0.001$) and prior endocrine sensitivity ($p < 0.001$) in 152 patients from the combined CDK-Validation-1 and CDK-Validation-2 cohorts. *P*-values (*p*) were determined by Fisher's exact test. **d** Kaplan-Meier curves of PFS (Log Rank $p < 0.001$; Cluster 3 vs Cluster 1 $p < 0.001$) (left) and OS (Log Rank $p < 0.001$; Cluster 3 vs. Cluster 1 $p = 0.004$) (right) of the 4 ctDNA-based clusters in 152 patients of the combined CDK-Validation-1 and CDK-Validation-2 cohorts. Source data are provided as a Source Data file.

metastasis), we analyzed the 150 DNA-based signatures across 158 tumor tissues (either primary tumors or metastatic tumors). The proportion of primary tumors falling in Clusters 1, 2, 3 and 4 was 47.6%, 27.0%, 11.1% and 14.3%, respectively. The proportion of metastatic tumors falling in Clusters 1, 2, 3 and 4 was 27.0%, 39.3%, 19.1% and 14.6%, respectively (Supplementary Fig. 23). Thus, an enrichment for cluster 1 (better prognosis) was observed in primary tumors compared to metastatic tumors ($p = 0.010$). This enrichment for a more aggressive tumor biology in the metastatic setting is consistent with previous studies[31,32].

## Discussion

Through in-silico supervised learning analysis of The Cancer Genome Atlas (TCGA) early-stage breast cancer dataset we previously identified 150 DNA copy-number-based multi-feature predictors able to track a variety of biological processes at the RNA and protein level[10]. Here, we addressed if our approach could be applicable in liquid biopsies (i.e., plasma ctDNA). In this report, we predict complex tumor phenotypes like tumor cell proliferation rates using blood samples from patients with metastatic cancer. Our discovery should lead to opportunities for biomarker discovery, validation, and implementation in oncology.

In metastatic breast cancer, phenotype-based classification using tumor tissue RNA expression profiling (i.e., Luminal A, Luminal B, HER2-enriched and Basal-like) is prognostic and might predict treatment benefit[5-7]. Specifically, both non-luminal subtypes within HR+/HER2- disease have been associated with a poor outcome in the context of endocrine-based treatment. In addition, the Basal-like subtype within HR+ disease is associated with a lack of benefit from endocrine therapy and CDK4/6 inhibition[5], a finding highly concordant with ours. Another example within metastatic HER2+ disease is the result of a phase II PATRICIA trial (NCT02448420), where the benefit of endocrine therapy and palbociclib seemed restricted to ~50% of patients with a Luminal A or B subtype[33]. Finally, in advanced TNBC, the identification of the Basal-like versus non-Basal-like subtypes revealed significant differences in response rates to carboplatin versus docetaxel in non-Basal-like disease in the TNT phase III trial[9]. Based on these findings, several prospective clinical trials are currently selecting patients based on their tumor's RNA-based phenotype (e.g., TATEN (NCT04251169), PATRICIA II (NCT02448420), NEREA (NCT04460430) and HARMONIA (NCT05207709)).

Among the different phenotypic features identified in ctDNA, the RB-LOH signature[16] was found as one of the top signatures associated with survival outcomes in patients with advanced HR+/HER2- breast cancer treated with endocrine therapy in combination with a CDK4/6 inhibitor. CDK4/6 kinases associate with cyclin D proteins during transition from G1 to S phase of the cell cycle[21]. The cyclin D/CDK4/6 complex phosphorylates RB, dissociating it from the E2F transcription factors, which are ultimately responsible for cell cycle progression[34]. In this scenario, loss of RB is an evident cause of resistance to CDK4/6 inhibitors[21], and various preclinical studies support this hypothesis[22,23]. In addition, mutations in *RB*, which are rare, are responsible for resistance to CDK4/6 inhibition in a few patients[24,25,35].

The RB-LOH mRNA-signature was originally developed by comparing gene expression between DNA assessed RB-LOH-positive breast tumors versus LOH-normal tumors[16]. Expression of 452 genes varied with RB-LOH status and 423 were highly expressed in tumors with RB-LOH. These genes were associated with cell cycle-related biological processes, including cell division, DNA metabolism, spindle organization and biogenesis and response to DNA damage, and many were known E2F-regualted genes[16]. If one closely examines the selected features of the DNA-copy number based RB-LOH signature, it includes 10s of features that encompass many chromosomal regions known to harbor unambiguous acknowledged players of the cell cycle including *RB1*, *E2F1,3*, *CCND1,2,3*, *CDK6* and *MYC*; thus an important quality of these multi-feature Elastic Net predictors is that they have been trained to 'sense' many parts of a given pathway and to combine these into one objective machine learning trained predictor, which links these together into a single quantitative score. As expected, the frequency of the RNA-based RB-LOH signature varied by intrinsic subtype with the lowest and highest LOH frequencies observed in the Luminal A (15%) and Basal-like (72%) subtypes, respectively, concordant with the known lack of RB function in Basal-like tumors and thus a lack of benefit of CDK4/6 inhibitors biology[5].

The ability to detect and monitor tumor phenotypes in blood samples opens opportunities for precision oncology. On one hand, tumor tissue biopsies of metastatic disease are difficult to perform, pose risks to the patient and do not fully capture the intra-patient tumor heterogeneity[36]. On the other hand, tumor biology evolves over time and the biology captured in a tissue sample might not fully recapitulate the biology present at the latter times decisions are often made in the metastatic setting. For example, ~1/3 of primary breast tumors switch molecular subtype when relapse occurs[31,32,37], a proportion that increases to ~50% if the primary tumor was Luminal A. In addition, large relative changes in tissue gene expression have been found between primary and metastatic breast cancer. For example, 47 of 105 breast cancer-related genes (45%) were found differentially expressed across 123 paired primary-metastatic tumors[31]. Interestingly, expression of proliferation-related genes was better at predicting OS in metastatic disease when analyzed in metastatic tissue rather than primary tissue[31]. This is concordant with our findings that the RB-LOH ctDNA-signature is better associated with prognosis when measured in ctDNA prior to initiating therapy compared to the same signature evaluated in archival tissue samples coming from the primary tumor, which typically predate the metastatic specimen by many years.

Our findings should facilitate the development of prospective trials in specific clinical scenarios selecting patients with a more homogenous tumor biology. For example, patients with HR+/HER2- advanced breast cancer and a high score in the ctDNA-based RB-LOH signature could be randomized to a chemotherapy-based treatment strategy versus the standard-of-care of endocrine therapy and CDK4/6 inhibition. In addition, tracking RB-LOH and other ctDNA-signatures might better help to understand why tumors progress and make treatment decisions based on that information. For example, a trial could evaluate endocrine therapy alone in patients with HR+/HER2-

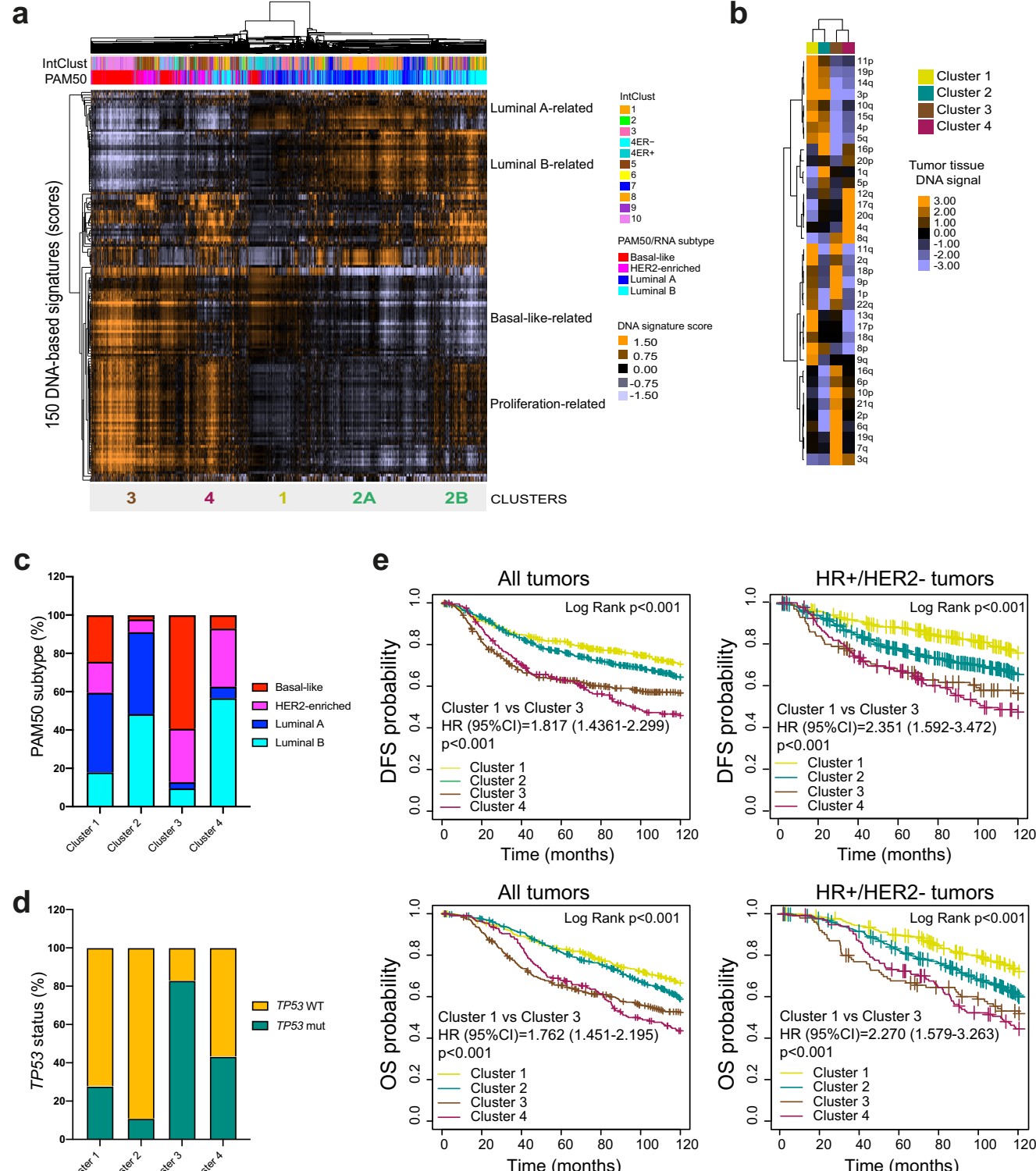

**Fig. 5 | DNA-based tumor profiles in tissue samples and association with clinical outcomes. a** Unsupervised cluster analysis of 1,689 tumor samples (columns) from METABRIC dataset[29] and the 150 DNA-based signatures scores (rows). Orange and violet colors represent scores above and below the median value of the signature across the dataset. Below the array tree, the InctClust classification[29] and the PAM50 molecular subtypes are shown for each sample. The predictor of the 4 clusters, which was derived from plasma ctDNA data in the Plasma-1 cohort (see Fig. 4), was applied in METABRIC and the results for each sample are shown below the data matrix. **b** Heatmap of signals from individual chromosomic regions found significantly associated with each cluster 1-4. Differentially represented segments were determined using a multi-class SAM analysis (FDR < 5%) **c** PAM50 molecular subtype distribution across the 4 DNA-based clusters in 1689 breast tumors of the METABRIC dataset[29]. **d** *TP53* mutation distribution across the 4 DNA-based clusters in 1,517 breast tumors of the METABRIC dataset[29]. **e** Kaplan-Meier curves of DFS (left) and OS (right) of the 4 DNA-based clusters assessed in all tumors (*n* = 1689) (DFS Log Rank *p* = 3.51e-10; Cluster 1 vs 3 *p* < 0.001; OS Log Rank *p* = 1.56e−09; Cluster 1 vs 3 *p* < 0.001) and HR+/HER2- tumors (*n* = 1131) (DFS Log Rank *p* < 0.001; Cluster 1 vs 3 *p* < 0.001; OS Log Rank *p* < 0.001; Cluster 1 vs 3 *p* < 0.001) of the METABRIC database[29]. Source data are provided as a Source Data file.

advanced breast cancer with a high score of the ctDNA-based Luminal A signature, and then add CDK4/6 inhibition when the ctDNA-based Luminal B signature increases during therapy versus initiating endocrine therapy and CDK4/6 inhibition upfront. Another clinical situation is when one is forced to change treatment after a single metastatic lesion progresses while the other metastatic lesions are radiologically under control and the patient is doing clinically well. In this context, detection of a low proliferation status in ctDNA might open opportunities such as considering local therapy approaches on the progressing lesion and continue with the same systemic therapy.

Our study has limitations. First, ~39% of patients had a TF < 3%, and ctDNA-signatures could not be evaluated. In addition, ~30% of patients had a TF of 3–10% and this might limit the detection of the ctDNA-based signatures. Further studies evaluating deeper ctDNA sequencing strategies and signature detection in patients with very low TF, including those with early-stage disease, is warranted. For example, expressed genes might be inferred by evaluating nucleosome footprints from whole-genome sequencing of plasma DNA[38]. Second, this was an exploratory study, and we did not use a sample size calculation but we used all the samples available for the correlative analysis. The lack of formal design through pre-planned analysis prohibits inference of negative results. Third, we did not address if any of the 150 signatures predict clinical outcome in advanced HER2+ or TNBC or benefit from other targeted therapies. Fourth, we decided to focus on signatures previously identified and validated. Fifth, although both independent datasets collected plasma samples prospectively from patients treated with endocrine therapy and a CDK4/6 inhibitor, the cohorts are not from clinical trials and are prone to potential biases such as patient selection, inconsistent evaluation of the disease during therapy and subjectivity in determining drug response and progressive events. Sixth, we did not compare the prognostic value of the ctDNA-based signatures measured at baseline with the prognostic value of early ctDNA dynamics after 2–4 weeks of initiating therapy[11,39]. Seventh, ctDNA-signatures might not capture well tumor immune-related biology as tumor tissue DNA-signatures.

To date, DNA sequencing has identified few FDA-approved actionable genetic alterations in cancer, especially breast cancers. Phenotypic characterization using multi-gene RNA-based expression might add biological and clinically relevant information. However, implementation of tumor-based RNA-based gene expression profiling in the metastatic setting is a major challenge since tumor tissue is often not readily available. Here, we demonstrate that complex tumor phenotypic traits can be identified in ctDNA and may provide clinical value. Our ctDNA-based multi-gene signature approach opens avenues for discovering and implementing biomarkers in oncology, especially in the advanced disease setting where liquid biopsies are becoming commonplace.

## Methods

The Hospital Clínic Barcelona institutional ethics committee approved the study in accordance with the principles of Good Clinical Practice, the Declaration of Helsinki, and other applicable local regulations (HCB/2019/0666). Written informed consent was obtained from all patients before enrolment. The medical records were retrospectively reviewed to obtain the necessary clinical data.

### Study participants and samples

We collected 246 plasma samples from 207 patients (Plasma-1 cohort) treated at Hospital Clinic of Barcelona ($n = 190$), Hospital 12 de Octubre in Madrid ($n = 15$) and Institut Català d'Oncologia ($n = 2$) at different stages of their metastatic disease: 209 plasmas from 179 patients with advanced HR+/HER2- breast cancer, 19 plasmas from 16 patients with HER2+ advanced breast cancer, 17 plasmas from 16 patients with advanced TNBC and 1 plasma from 1 patient with unknown ER and HER2 status (Fig. 1a). Plasma-1 cohort included 124 baseline pre-

treatment blood plasma samples from 124 patients with HR+/HER2- advanced breast cancer treated with endocrine therapy in combination with a CDK4/6 inhibitor (i.e., palbociclib, ribociclib or abemaciclib) between the years of 2018 and 2021 (CDK-Validation-1 cohort). All plasma samples were obtained before the start of treatment (Fig. 1b). In 7 patients, we obtained an additional plasma sample after progressing while on therapy.

Additionally, we included a second independent cohort of 357 plasma samples of patients with advanced breast cancer (Plasma-2 cohort) (Fig. 1A), including 121 baseline samples of patients with HR+/HER2- advanced breast cancer treated with endocrine therapy in combination with a CDK4/6 inhibitor (i.e., palbociclib, ribociclib or abemaciclib) at Hospital Clinic of Barcelona ($n = 50$) and Vall d'Hebrón Institute of Oncology ($n = 71$) (CDK-Validation-2) (Fig. 1b). Only baseline pre-treatment plasma samples were available from this cohort. Moreover, we analyzed plasma samples from 14 healthy individuals.

In addition, 185 FFPE tumor tissues from patients treated at Hospital Clinic of Barcelona were collected, including samples from 110 patients with available plasma samples (in Plasma-1 cohort), and samples from 17 patients with HR+/HER2- advanced breast cancer who did not have plasma samples but were treated with endocrine therapy and a CDK4/6 inhibitor.

Publicly available DNA, RNA and clinical data from METABRIC and the MSKCC datasets were obtained from cBioportal.

### DNA-sequencing of plasma samples

Approximately 30 mL of peripheral blood was collected into K2-EDTA Vacutainer tubes (Becton Dickinson) and plasma isolation was performed within 2 h of blood collection through two centrifugation steps. Centrifugation at 1600 x g for 10 min at 4 °C separated plasma from peripheral-blood cells. Approximately 12 mL of plasma were obtained per patient, which were subsequently centrifuged at 16,000 x g for 10 min at 4 °C to remove the residual supernatant and any remaining contaminants including cells. Plasma samples were then aliquoted in 1.5 mL tubes and immediately stored at −80 °C. cfDNA was obtained from 3 mL of plasma using the QIAamp Circulating Nucleic Acid Kit (QIAGEN Inc.) according to the manufacturer's instructions and quantified with a Qubit dsDNA high-sensitivity assay kit and the Qubit 4.0 fluorometer (Life Technologies, Carlsbad, CA, USA). cfDNA was concentrated using SpeedVac to fulfil the requirements for library preparation. Library preparation was performed by ligating unique dual indexes (UDI) custom adapters to a minimum of 10 ng of the isolated cfDNA (10–50 ng dsDNA). More specifically, the fragment ends of cfDNA were blunted and 5′ phosphorylated and, after that, 3′ ends were A-tailed to favour adapter ligation. Adapters were 10 bp – UDI as recommended to mitigate errors introduced by index-hopping or switching in Illumina instruments with patterned flow cells, such as the NovaSeq 6000. Indexed libraries were quantified by qPCR using the KAPA Library Quantification Kit (Roche Sequencing Solutions), pooled, and sequenced in a NovaSeq 6000 Illumina at 0.5X mean coverage with read length of 2 ×150 bp. ShWGS was analyzed with hmmcopy_utils (https://github.com/shahcompbio/hmmcopy_utils) and ichorCNA v0.2.0 (https://github.com/broadinstitute/ichorCNA), with a bin size of 500 kb and default parameters[12]. ichorCNA is a previously reported tool by Adalsteinsson et al.[12] for estimating the fraction of tumor in cfDNA from ultra-low-pass WGS. ichorCNA uses a probabilistic model, implemented as a hidden Markov model, to simultaneously segment the genome, predict large-scale copy number alterations, and estimate the TF of an ultra-low-pass whole genome sequencing samples. ichorCNA is optimized for low coverage (-0.1X) sequencing of samples and has been benchmarked using patient and healthy donor cfDNA samples. Adalsteinsson et al. reported that, using a TF cutoff of 3%, ichorCNA achieves a sensitivity of 0.95 for detecting presence of tumor and a specificity of 0.91 for correctly classifying a healthy donor[12].

## Analytical validation analysis of the methodological approach

To validate the approach, the following analytical analyses were performed. First, since the coverage used in this study was at the higher end of what is considered ultra-low pass WGS by definition (0.1–0.5X coverage), we performed an in-silico downsampling of sample coverage for 12 samples in order to assess performance at different coverage. Different TFs were represented within these 12 samples. For each sample, a subset of aligned reads was selected to account for: 2X (40 M reads, 151 nucleotide [nt] long), 1X (20 M reads, 151 nt long), 0.5X (10 M reads, 151 nt long) and 0.1X (2 M reads, 151 nt long) coverages. Reported TFs were consistent across coverages (Supplementary Data 12).

Second, correlation analyses of the bin-to-bin log2 values of these samples indicated a decrease in correlation of log2 values reported by iChorCNA for the 0.1X condition in samples of TF < 10%, although 0.5X (the coverage used in the presented study) performs similarly to 1X (Supplementary Data 13). Of note, the 0.5X coverage samples raise similar signature score values to 1X and 2X. For TFs below 20%, 0.1X coverage may raise different scores and is not recommended for this approach. Moreover, we applied our CNA signatures on the 11 samples with TF ≥3%. Again, 0.5X coverages raise similar profiles across all signatures to 1X and 2x (Supplementary Data 14). As an example, RB-LOH signature score using different coverage is shown in Supplementary Fig. 24.

Third, regarding determining the sensitivity of each signature according to TF, we performed 'serial' dilutions of TF by in-silico mixing reads from the 6 cases with TF > 50% with reads from the pooled 14 healthy control samples to generate TFs of: 50%, 20%, 10%, 5%, and 1%. The resulting samples were analyzed with iChorCNA, and CNA signatures run on each (Supplementary Data 15). At 1% TFs, iChorCNA may fail to detect tumor profiles in the shWGS data (5/6 fails at 1% in-silico TF). According to our data, 5–10% TF has an acceptable overall failure rate (<20%). With regards to applying the CNA signatures, there is a good correlation between samples >10% TF. As an example, RB-LOH signature score is shown in Supplementary Fig. 25.

Next, we evaluated, from a prognostic perspective, the effect of removing an increasing number of segments of a determined signature. For instance, we randomly removed 20, 40, 80, or 160 DNA-based segments of the RB-LOH signature (measured in plasma) and evaluated its association with PFS. We repeated this analysis 500 times and estimated the average hazard ratio and $p$. Removing features of the RB-LOH signature affected its prognostic ability (i.e., hazard ratio decreased, and p increased) (Supplementary Fig. 26).

Finally, shWGS was performed in 14 healthy individuals, and the log2 values by ichorCNA estimates were determined to be on average 0 (Supplementary Data 16).

## DNA-sequencing of FFPE tumor samples

DNA obtained from FFPE-derived tissues was purified with the QIAamp DNA FFPE Tissue kit (QIAGEN Inc.) for all samples available, following manufacturer's instructions. Quantification was performed with a Qubit dsDNA broad-range assay kit and the Qubit 4.0 fluorometer (Life Technologies, Carlsbad, CA, USA). A minimum of 100 ng of extracted DNA was processed for library preparation using a custom hybridization-based capture panel targeting 435 genes with reported somatic mutations in different tumor types (VHIO-300 v4 panel) performed with Agilent SureSelectXT Low Input Target Enrichment System (Agilent Technologies, Inc). Indexed libraries were quantified by qPCR using the KAPA Library Quantification Kit (Roche Sequencing Solutions), pooled, and sequenced in a HiSeq 2500 Illumina (2 x 100 bp) at an average coverage of 500X. Reads were aligned to the hg19 reference genome with BWA[40], applied GATK[41] base quality score recalibration, indel realignment, and duplicate removal. Variant calling (VarScan2 v2.4.3) required a minimum of 7 reads supporting the variant allele to call a mutation. The sensitivity of the technique is 5% Minor allele frequency (MAF) for Single nucleotide variant (SNVs) and

10% MAF for INDELs. Frequent single nucleotide polymorphisms (SNPs) in the population were removed based on the gnomAD database (allele frequency ≤0.0001). CNA were calculated from an in-built genome-wide SNP backbone targeting 20000 SNPs using CNVkit (v0.9.6.dev). Data was manually curated, and classification of identified variants was performed using publicly available databases (COSMIC, cBioPortal, ClinVar, VarSome, OncoKB).

## DNA-based signature estimation

For both tumor DNA sequencing and plasma cell-free ctDNA sequencing, segmentation files from CNVkit output (for tumor DNA) and ichorCNA output (ctDNA) were first mapped to gene-level feature. Values from 514 DNA segments were then determined as described in Xia et al.[10]. Briefly, each segment score was calculated as the mean copy number score across genes within the segment. The coefficients of DNA segments for predicting gene signatures were obtained from Xia et al. DNA-based signature scores were calculated as the weighted average of DNA segment values for each sample.

For ctDNA, TF and tumor ploidy were estimated by ichorCNA. For ctDNA samples with TF > 0, TF and tumor ploidy adjusted signature scores were calculated by first adjusting copy number values in ichorCNA segmentation file: adjusted_copy_number_ratio = log2(logR_copy_number/tumor_ploidy). Then DNA-based signature scores were derived the same as described for tumor tissue. For calculating the number of altered segments, we used arbitrary gain/loss threshold of +/− 0.07 for unadjusted segment values and 0.32/−0.42 for adjusted segment values[10]. Segments with values above the gain threshold or below the loss threshold were called altered.

## Gene expression analysis of FFPE tumor samples

RNA was extracted using the High Pure FFPET RNA isolation kit (Roche, Indianapolis, IN, USA) following manufacturer's protocol. One to five 10-µm FFPE slides depending on tumor cellularity were used for each tumor sample, and macrodissection was performed, when needed, to avoid normal tissue contamination. A minimum of ~100 ng of total RNA was analyzed on the nCounter platform[42] (Nanostring Technologies, Seattle, USA) using the 770-gene Breast Cancer 360™ Gene Panel, which includes the 50 PAM50 genes. Gene expression for each sample was independently normalized to the geometric mean of 5 housekeeping (*ACTB, MRPL19, PSMC4, RPLPO,* and *SF3A1*), and research-based PAM50 subtyping was determined[15].

## METABRIC and MSK breast cancer datasets

Clinical-pathological data was obtained from cbioportal[43]. Processed DNA segment values were downloaded, and DNA-based signature scores were calculated as the weighted average of DNA segment values for each sample. Treatment and clinical outcome information from the MSK dataset was obtained from Table S2 from Razavi et al.[26].

## A DNA-based 4 subtype predictor

To identify the 4 subtype clusters using DNA-based data, we selected signatures that were significantly differentially expressed across the 4 clusters identified in ctDNA using a multi-class significance analysis of microarrays (SAM)[44] with <5% FDR. Then we used the selected gene list and calculated 4 centroids from the training data. For every new sample in METABRIC[29], we calculated the Euclidean distances to the 4 centroids and assigned a cluster class to each sample based on the nearest centroid.

## General statistical procedures

Categorical variables were expressed as number (%) and compared by $\chi^2$ test or Fisher's exact test. Differentially expressed signatures between two groups were identified using a two-class unpaired SAM with an FDR < 5%. Differentially expressed signatures between two timepoints (i.e., baseline versus post-progression to endocrine therapy

and a CDK4/6 inhibitor) were identified using a two-class paired SAM with an FDR < 5%. Estimates of survival were from the Kaplan–Meier curves and tests of differences by the log-rank test. Univariate and multivariable Cox models for PFS and OS were used to test the prognostic significance of each variable. The Bonferroni correction method was used to control the family-wise error rate in case of multiple comparisons. PFS was defined as the period from initiation of endocrine therapy and a CDK4/6 inhibitor until disease progression or date of last follow-up. OS was defined as the period from initiation of endocrine therapy and a CDK4/6 inhibitor until death or date of last follow-up. All cluster analyses were displayed using Java Treeview version 1.1.3. Average linkage hierarchical clustering was performed using Cluster v3.0[45]. Two-sided $p$ < 0.05 were considered statistically significant. Statistical computations were carried out in R 4.0.3 (http://cran.r-project.org).

### Reporting summary

Further information on research design is available in the Nature Portfolio Reporting Summary linked to this article.

## Data availability

The data collected for the study are not available, as participants of this study did not agree for their data to be shared publicly. However, we encourage investigators interested in data access and collaboration to contact the corresponding author (Aleix Prat, alprat@clinic.cat). Access can be obtained for academic use only under a data transfer agreement and upon Ethics Committee approval. The timescale for this process is approximately 6 months and the data will be available for 3 years. The data generated in this study and presented in the figures are provided in the Supplementary Data/Source Data files. Source data are provided with this paper.

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

## Acknowledgements

Instituto de Salud Carlos III PI19/01846 to AP, Breast Cancer Now 2018NOVPCC1294 to AP, Fundació La Marató TV3 201935-30 to AP, RESCUER, funded by European Union's Horizon 2020 Research and Innovation Programme under Grant Agreement No. 847912 to AP, and Fundación CRIS contra el cáncer PR_EX_2021-14 to AP. Fundación científica AECC Ayudas Investigador AECC 2021 INVES21943BRAS to FBM.

## Author contributions

A.P., A.V., F.B.M., C.M.P., and J.S.P. designed the study. A.P., A.V., F.B.M., O.M.S., E.S., M.B., P.G., D.M., T.P., A.R., N.C., B.A., M.V., M.M., E.B., M.O., E.F., J.M., R.S.B., A.S., C.S., E.C., P.T., M.M., B.G.F., and P.V. contributed to data collection and assembly. A.P., A.V., C.M.P., J.S.P., F.B.M., Y.X., L.P., and M.G.R., interpreted and analyzed the data. All authors wrote and reviewed the report and approved the final version for submission.

## Funding

The study was designed and performed by investigators from Reveal Genomics, Hospital Clinic/IDIBAPS and VHIO. All authors had full access to all data in the study and had final responsibility for the decision to submit for publication.

## Competing interests

Potential conflicts of interest are the following: A.P. reports advisory and consulting fees from Roche, Pfizer, Novartis, Amgen, BMS, Puma, Oncolytics Biotech, MSD, Guardan Health, Peptomyc and Lilly, lecture fees from Roche, Pfizer, Novartis, Amgen, BMS, Nanostring Technologies and Daiichi Sankyo, institutional financial interests from Boehringer, Novartis, Roche, Nanostring, Sysmex Europa GmbH, Medica Scientia inno. Research, SL, Celgene, Astellas and Pfizer; stockholder and consultant of Reveal Genomics, SL; a patent PCT/EP2016/080056 and the DNADX patent filed (EP22382387.3). F.B-M. has the DNADX patent filed (EP22382387.3). P.V. has the DNADX patent filed (EP22382387.3). C.M.P is an equity stockholder and consultant of BioClassifier LLC, and for Reveal Genomics. C.M.P is also listed as an inventor on patent applications for the Breast PAM50 assay, and a patent application on DNA-based predictors of breast tumor phenotypes. J.S.P is an equity stockholder and consultant for Reveal Genomics and is also listed as an inventor on patent applications for the Breast PAM50 assay, and a patent application on DNA-based predictors of breast tumor phenotypes. A.V. has the DNADX patent filed (EP22382387.3). The remaining authors declare no competing interests.

## Additional information

[1]Translational Genomics and Targeted Therapies in Solid Tumors, August Pi i Sunyer Biomedical Research Institute (IDIBAPS), Barcelona, Spain. [2]Reveal
Genomics, Barcelona, Spain. [3]Department of Medical Oncology, Hospital Clinic of Barcelona, Barcelona, Spain. [4]Department of Medicine, University of
Barcelona, Barcelona, Spain. [5]Institute of Oncology (IOB)-Hospital Quirónsalud, Barcelona, Spain. [6]SOLTI cooperative group, Barcelona, Spain. [7]Department
of Pathology, Hospital Clinic de Barcelona, Barcelona, Spain. [8]Department of Medical Oncology, Vall d'Hebron University Hospital, Autonomous University of
Barcelona, Barcelona, Spain. [9]Breast Cancer and Melanoma Group, Vall d'Hebron Institute of Oncology, Barcelona, Spain. [10]Lineberger Comprehensive
Cancer Center, University of North Carolina, Chapel Hill, USA. [11]Department of Medical Oncology, Institut Catalan of Oncology - Badalona, Hospital Germans
Trias i Pujol, Badalona, Spain. [12]AIDS Research Institute-IrsiCaixa and Health Research Institute Germans Trias i Pujol (IGTP), Hospital Germans Trias i Pujol,
Badalona, Spain. [13]Cancer Genomics Group, Vall d´Hebron Institute of Oncology (VHIO), Barcelona, Spain. [14]Department of Medical Oncology, Hospital
Universitario 12 de Octubre, Madrid, Spain. [15]Department of Genetics, University of North Carolina, Chapel Hill, USA. [16]These authors contributed equally:
Aleix Prat, Fara Brasó-Maristany, Olga Martínez-Sáez. ✉e-mail: alprat@clinic.cat

