## [Peer Review File · Nature Communications]

Circulating Tumor DNA Reveals Complex Biological Features with Clinical Relevance in Metastatic Breast CancerREVIEWER COMMENTS

Reviewer #1 (Remarks to the Author): expert in breast cancer genomics

The article by Prat, Brasó-Maristany, and Martínez-Sáez utilizes a previously described signature derived from supervised learning integrative computational approach to predict RNA-based tumor expression signatures using data from DNA copy-number alterations. This method was developed by the same group (in the large sense, Perou et al) and understanding that method is essential to understand and appreciate the presented results here.

In the present manuscript the authors attempt a remarkable thing: because the expression based classification can be deduced from copy number alterations (again, one needs to read ref 10 to be convinced), then if one can detect the copy number alterations in ctDNA in the plasma/serum, then one can classify tumors non-invasively from a plasma sample. This is exactly what the authors show for ESR1+ and Her2 patients, and it is in my mind, as just said, remarkable.

The authors analysed ctDNA from 246 samples with shallow whole genome DNA sequencing. A majority of the patients were ER+/Her2- (207), but some from the other subtypes were also included (Her2+ and TNBC, 16 patients in each). Also samples for some patients were available at different time points, so number of plasma samples is not the same as number of patients, then at some point the analysis is restricted to those who have both tumor and plasma sample (54), and before and after treatment (124), and only those with TF>3% were further analysed, so this ensemble of numbers may be described better.

Of the 246, samples 178 (72.4%) had a Tumor cell Fraction (TF) of $\geq 3\%$ (range 156 4-84%; median 9.4%), according to the ichorCNA tool. Here one needs to know why TF of 3% was chosen as a cut of? How does the classification deteriorate below this threshold? Patients with (TF) of <3% do they have better prognosis?

Then the authors could ask the questions:

-of 150 detectable signatures described in the tumor how many do we detect in plasma?
I could not find this information explicitly provided. Could have missed it. One should be able to see this perhaps by comparing the number of rows in the heatmaps in figure 3 and 4.

-Are some of these signatures over/underrepresented in plasma compared to tumor?

-What is the correlation of each signature to tumor burden and TF.

Here the authors present the results unadjusted ctDNA scores (TF correlates with overall number of detected CNA alterations and with 46 of 150 (31.0%) ctDNA-based signature scores), and adjusted for TF ctDNA-based signature scores (here the correlation to TF of course disappears, this gives a possibility for a more unbiased (byTF) comparison of ctDNA signatures from patient to patient.

Then it is a bit unclear, do the authors continue the analyses (from line 177 on) with the adjusted or not adjusted ctDNA scores? F.ex. in the case of the comparison of tumor burden (as number of met sites) to TF to ctDNA scores? Because the ctDNA-based signatures that highly correlated with TF represented relevant biological processes, reducing their effect in the posterior analyses with adjusted data may be unfair?

A moderate correlation between tumor and plasma is observed for around 40 ctDNA signatures, increasing to 60 if the timepoint of sample taking is closer. One could highlight some of those in the text as well say if the 60 contain the 40. It is readable from comparing the suppl tables, but still.

Comments to figures:

Figure 1.

- 1A. Can take the space of signature 1, signature 2, signature 3...to instead illustrate all the compared pairs (tumor-plasma, before after treatment)
- 1B. On the X axis add (TF) to Tumor fraction (TF) in order to signify that it is the same thing that is adjusted for in the right panel (one may get confused and think it is pathologically assessed tumor fraction in the tumor). Also a brief mention about how TF is calculated in the legend may help in addition to the Method section.
- 1C. X axis, I would replace "tissue based" with "tumorDNA-based"

Figure2.

D and E should be swapped, both to accommodate the left to right reading pattern, and also because the hazard ratio information follows more naturally the survival curves.

Figure D (which I suggest should become E) requires specific attention, as the columns need to be better explained. Of "Average ctDNA signal of 16 features of the RB-LOH DNA-based signature (column on the left), weight and direction of each feature (column in the middle) in the original signature as reported in Xia et al.(13) and mean change of the 16 features (column on the right) in 7 patients with paired plasma samples only the third is clear to the reader. How these values are obtained and what they mean should be explained in the legend instead of referring the reader to another paper. "original tissue based weight" cannot say much to one who would like to understand what was done just from the figure.

Figures 3 and 4 are, in my mind, somewhat less central to the present report. Would this subclassification based on ctDNA from plasma samples from this particular set of patients and its correlation to PAM50 and in fact be clinically relevant? What more does it say in addition to what was already estimated by the direct tumor-plasma correlation of ctDNA pathways? Maybe good to have. The data in Figure 4- was it not the main topic of the much referred to reference 10? I mention that just because a lot of attention also in the text is given to describe these results, which seem to me less central to the very important message this paper brings.

Reviewer #2 (Remarks to the Author): clinical expertise in breast cancer and CDK4/6 inhibitor response

Analysis of ctDNA is a valuable tool that has proven to be useful in identifying mutations and copy number alterations for targeted therapy. Here, the authors apply machine learning model, previously developed using tissue DNA, to detect 150 multi-gene signatures in ctDNA with the aim of identifying biologic features of the cancer including measures of estrogen receptor signaling and tumor proliferation. They examined a data set of 246 plasma samples, mostly from patients with advanced HR+/HER2- breast cancer, but included only those that had a tumor cell fraction $\geq 3\%$ (178 samples). Using a subgroup of 54 patients with paired ctDNA and tumor DNA, they found very limited correlation (0.4 average) between tumor and plasma signatures, raising the questions of what is really being measured by the ctDNA signatures if it is not reflective of the tumor tissue signatures, which is not addressed by the authors. The authors provided evidence supporting correlations between ctDNA proliferation signatures and tumor cell fraction, luminal A signatures and bone-only disease, luminal signatures and ER+ status, and HER2 signatures with HER2+ disease. Additionally, using a set of 87 pre-treatment plasma samples, they identified ctDNA signatures associated with poor prognosis and response to treatment with endocrine therapy and CDK4/6 inhibition, and focused on RB-LOH signature. Using an independent cohort, which after filtering to tumor fraction $> 3\%$ was only 65 patients, they found that RB-LOH signature in ctDNA was associated with worse PFS. They also examined a cohort from MSK and found similar association between RB-LOH in tumor tissue taken < 1 year from starting therapy and PFS – note that this was signatures derived from tumor tissue and ctDNA signature was not reported for the MSK set. Additionally they performed unsupervised clustering on ctDNA signatures and found 4 clusters that were associated with PFS and OS in both ctDNA validation cohort and METABRIC tumor tissue cohort. Overall the authors conclude that applying their machine learning model to identify 150 signatures in ctDNA provides additional information about disease biology.

Major Comments:

- 1) The abstract needs to be restructured to reflect the methods and data presented in the paper, it is now very general and vague.

- 2) The authors note initial signatures were developed as per Reference #10. It is the understanding of this reviewer that the initial signatures were developed from TCGA data, which include early stage tumors. The current paper focuses on advanced breast tumors. The signatures in advanced tumors may differ from those seen in early disease. One example is ESR1 mutations, rarely seen in early tumors, but commonly emerge in patients treated with endocrine therapy.
- 3) The paper is very dense and contains vast amount of data including many supplementary files. The authors should state their specific hypotheses and consider organizing the results and methods accordingly. They may also consider splitting the paper into two separate ones.
- 4) The descriptions of the cohorts studied in this paper are not clear. Consider assigning a number or letter to each cohort and use these consistently throughout the manuscript.
- 5) Consider adding a table describing each of the cohorts, including prior treatment, and response to treatment. The knowledge of prior treatment is important as the dynamics of the markers may change over time.
- 6) It appears that sample collection was prospective, but additional patient, tumor, and treatment information was collected from the patients' medical records. Please discuss possible bias.
- 7) Sample size considerations and power for each analysis are not included. Are the cohorts simply samples of convenience? Further explanations are required to understand the true level of evidence of each of the reports.
- 8) Please provide p-values for the correlations between tissue-based signatures and ctDNA-based signatures (page 6, starting line 195)? Specifically:
 - a. The correlation coefficients, even for samples obtained within 8 weeks, are pretty low, but hard to interpret as no p-values given except in the 2 examples in Figure 1C. If there is no, or only limited, correlation between the tissue signature and the ctDNA signature, then what exactly is being measured in the ctDNA? These data should be presented first, and needs to be explained, because all the other correlations reported could be irrelevant if the signatures detected in ctDNA are not reflective of signatures obtained from tumor tissue.
 - b. Later in the paper, the authors highlight the correlation between 48 features of tissue based RB-LOH signature and ctDNA, suggesting that correlation is between tissue and ctDNA is important (page 8).
 - c. On page 13, starting on line 390, the authors describe the correlation of 6 PAM50 RNA-based tissue signatures with each of the 150 ctDNA-based signatures, as well as correlation with gene expression in the n Counter Breast Cancer 360 Panel. This needs to be moved up in the paper and should be discussed along with relationship between ctDNA and tissue DNA signatures.
 - d. What is the association between tissue based signatures and the other features presented (i.e. bone only disease, tumor burden, ER status, HER2 status, response to endocrine therapy, etc)? Are tissue based signatures or ctDNA based signatures better associated with these features? It is mentioned on line 304 that RB-LOH ctDNA signature was associated with PFS but not RB-LOH tumor signature, suggesting that ctDNA based signature may be better, but what about the other signatures?
 - e. The authors comment in the introduction "the type of metastatic organ or site can compromise the expression patterns obtained from bulk RNA and might not reflect the intra-patient tumor heterogeneity" (p4, line 125) – perhaps this is the cause of the lack of correlation with tissue? It would be helpful if authors included the location of tissue biopsy specimen for metastatic patients and assessed correlation by metastatic site. What is the impact of the different sequencing methods used (shallow WGS for ctDNA vs targeted panel sequencing for tissue)? How dependent on sequencing method is the machine learning model? The overall lack of correlation between tissue and ctDNA signatures needs to be investigated further/better explained before anything else in the paper can be interpreted.
- 9) The authors state that "The observed correlation between proliferation-related ctDNA-based biology with TF suggested that TF reflects the biological aggressiveness of the disease beyond the patient's tumor burden" (page 6, line 177). This statement is not supported by the evidence provided. No evidence is given regarding the biological aggressiveness, such as rate of tumor growth, or rapidity of clinical progression. There is no support given for decision to use <3 versus ≥ 3 metastatic sites as a measure of tumor burden and I do not believe this is an accurate measure of tumor burden. Furthermore, method used to count metastatic sites was not provided (are 2 spots in the liver 2 sites or 1? How large does a spot need to be to be counted? – this needs to be included in the methods section). Imaging based estimates of volume of disease would be more meaningful if available. Otherwise, I would not refer to this as "tumor burden" in this paragraph, but rather as number of metastatic sites. The lack of correlation between proliferation-

related signatures and "tumor burden" could in fact reflect conclusions other than those given by the author including 1) proliferation signatures are not good markers of tumor burden, or 2) tumor burden is not well measured by greater or less than 3 metastatic sites. The correlation presented between luminal A signatures and bone only disease is more compelling. If the authors are not able to better support their claims regarding tumor burden, then the findings related to tumor burden should be omitted or else very carefully stated with much more limited conclusions.

10) Why did the authors focus on the RB-LOH signature? This was not the signature with the lowest p-value or highest HR. Without an explanation regarding why this signature was picked to focus on, it seems like cherry-picking. There is some discussion of relevance of RB-LOH in the discussion section, some of this should be included in the results to justify why the authors focused on this.

11) What is the association between ctDNA-based tumor subtypes (clusters 1-4, page 12) and clinical factors such as prior treatment, response to treatment, location of metastatic disease, etc? What, if any, is the clinical relevance of identifying these clusters? I found this discussed on page 14, beginning at li

ne 417. This should be reorganized so the discussion of the clusters goes with the clinical meaning (PFS, OS) of the clusters.

12) Given some of the comments listed above, it appears that the results are still hypothesis generating and the conclusions should be revised accordingly.

Minor Comments:

1) The authors characterize solid tumor biopsy in the metastatic setting as "challenging." Depending on location, biopsy of metastases can actually be relatively straight forward. I agree that liquid biopsy has multiple advantages over solid tissue biopsy, including less invasiveness to the patient. I advise the authors to adjust their language around solid tumor biopsy to be more specific regarding the perceived disadvantages vs advantages of liquid biopsy.

2) In line 546, the author state "To date, DNA sequencing has identified few FDA-approved actionable genetic alterations in cancer, especially breast cancers" and then go on to discuss that RNA expression phenotypes are associated with treatment benefit, implying that that expression signatures are more actionable. This is not supported by the literature. This sentence should be omitted or re-written.

3) Figure 4 legend suggests that unsupervised cluster analysis was performed on the METABRIC tumor sample set. My reading of the paper was that cluster analysis was performed on the ctDNA data set, fixed, and then applied to the METABRIC dataset. This should be clarified.

Reviewer #3 (Remarks to the Author): expertise in ctDNA profiling in breast cancer

Prat and colleagues propose to use ctDNA profiling for (noninvasively) measuring several biological features in the metastatic breast cancer. Building this upon their previous work, they use copy number alterations to estimate gene expression activity of 150 signatures. Due to low levels of ctDNA in early-stage breast cancer, the focus of the paper is on metastatic setting, where in most cases, (tumor) allele fractions are expected to be at least in single digits.

Cell-free DNA profiling is done by shallow whole genome sequencing (~0.5X). The idea of using copy number patterns to capture tumor phenotypes is very interesting and could be valuable in metastatic settings. This would be particularly useful due to its very low cost.

Major comments:

1- The authors use ichorCNA to evaluate tumor fractions. Unfortunately, while ichorCNA has been used in several studies so far, its performance is not well established (for both TF estimation & CNV detection); and therefore, before applying to the data generated in this study, one would need to know its sensitivity for CNV detection as a function of ctDNA conc. & sequencing coverage. Particularly, I would like to see whether ichor estimates (1) correlate well with SNV-based TFs, and (2) are sufficiently close to zero for non-cancer controls.

2- Can authors comment on the TFs in this study? They seem too high (about 25-30% higher than 20%). Is that comparable with other studies? A comparison with existing data would be valuable.

3- Is there any benefit in increasing sequencing depth? I am a bit skeptical that this method would work with 0.5X coverage (i.e., ultra low pass WGS), for ctDNA fractions less than 20%, especially

since individual copy number (segment) states are fed to (pre-trained) models to calculate the signatures. Some analytical LOD analyses are needed, e.g., sensitivity of each signature vs coverage/TF.

4- Figure 1B does not convey any additional information. The TFs are estimated by the copy number events and therefore correlation is clearly expected. Is there any correlation between number of CNV events and an orthogonal estimate of tumor fraction? Like SNVs?

5- Figure 2E. This analysis is confusing. Does adjustment for TF mean a multivariable analysis with both 'RB-LOH signature' and 'TF' in the survival analysis? Also, given that there are 224 copy number segments contributing to that signature, a significance analysis is needed to show that if they randomly select 224 copy number segments, the survival stratification would no longer exist, or at least not as strong.

6- The subtype predictor to generalize the four clusters from plasma ctDNA to tumor DNA is not properly evaluated. Authors should perform cross-validation to evaluate the performance of their proposed approach and then apply that as a classifier.

7- An analysis comparing CNV events in tumor vs plasma cfDNA is needed; perhaps similar to the one in Herberts et al (Nat 2022- Fig 3d).

8- Is the prognostic value of clusters in figure 3 more than TFs? How about total genome instability? Figure S15 shows that patients in cluster 1 have the smallest TFs, and clusters 3 & 4 have the largest. So, given the data in figure 4, this may indicate that this is about genomic instability and not necessarily the clusters found using the estimated signatures. It would be great to test this.

9- Why are overall survivals so different between figure 3 & 4? In fact, patients in clusters 3 & 4 have very poor outcome in the first two years compared with those in the same clusters in figure 4 (HR+/HER2- tumors).

10- How do expression inference methods, such as those proposed by Ulz et al (2016, 2019) or Esfahani et al (2022) work here? I would imagine one may group genes to estimate the signature activity from shWGS. I see that there is one sentence in the discussion about it, however I believe a comparison is needed to justify using CNVs and not the epigenetic footprints.

RESPONSE TO REVIEWERS

Reviewer #1 (Remarks to the Author): expert in breast cancer genomics

The article by Prat, Brasó-Maristany, and Martínez-Sáez utilizes a previously described signature derived from supervised learning integrative computational approach to predict RNA-based tumor expression signatures using data from DNA copy-number alterations. This method was developed by the same group (in the large sense, Perou et al) and understanding that method is essential to understand and appreciate the presented results here.

In the present manuscript the authors attempt a remarkable thing: because the expression-based classification can be deduced from copy number alterations (again, one needs to read ref 10 to be convinced), then if one can detect the copy number alterations in ctDNA in the plasma/serum, then one can classify tumors non-invasively from a plasma sample. This is exactly what the authors show for ESR1+ and Her2 patients, and it is in my mind, as just said, remarkable.

We thank the reviewer for this very positive comment. We believe our novel approach in plasma will allow the identification (and tracking) of key biology in patients with metastatic breast cancer beyond single gene alterations. In addition, our approach could be applied to other cancer-types. Our current plan is to standardize the assay from both tissue and plasma and seek prospective validation. Several companies and academic groups are interested in our technology/approach, and we plan to start 1-2 prospective trials in the upcoming 1-2 years.

The authors analysed ctDNA from 246 samples with shallow whole genome DNA sequencing. A majority of the patients were ER+/Her2- (207), but some from the other subtypes were also included (Her2+ and TNBC, 16 patients in each). Also samples for some patients were available at different time points, so number of plasma samples is not the same as number of patients, then at some point the analysis is restricted to those who have both tumor and plasma sample (54), and before and after treatment (124), and only those with TF>3% were further analysed, so this ensemble of numbers may be described better.

We thank the reviewer for this comment. We have now better clarified the number of patients/samples in each cohort. To do so, we have created a new Figure (now Figure 1), which has all the information about the different plasma cohorts analyzed, the number of patients/samples, the proportion of HR+/HER2-, HER2+ and TNBC cases, the cases with tumor tissue DNA sequencing data, the number of cases with RNA expression data and the proportion of samples with a TF above 3%.

Of the 246, samples 178 (72.4%) had a Tumor cell Fraction (TF) of $\geq 3\%$ (range 156 4-84%; median 9.4%), according to the ichorCNA tool. Here one needs to know why TF of 3% was chosen as a cut of? How does the classification deteriorate below this threshold? Patients with (TF) of $< 3\%$ do they have better prognosis?

We thank the reviewer for pointing this out. The reason of choosing a TF cutoff of 3% is based on the original ichorCNA study by Adalsteinsson and colleagues (Nature Communications, 2017), where they showed that a TF of 3% achieves a sensitivity of 0.95 for detecting presence of tumor and a specificity of 0.91 for correctly classifying a healthy donor. We have now added this justification in the methods section of the manuscript titled "*DNA-sequencing of plasma samples*".

Regarding how the signatures were detected below the 3% cutoff, we did not pursue this since no detection of altered ctDNA segments was found (only 1 case of 246 cases, 0.4%). In our plasma-1 dataset (n=178), the median number of altered segments in cases with a TF of 3-10%, 10-20% and >20% was 51.5 (range 5-389), 240.5 (range 9-429) and 229 (range 29-433), respectively.

Finally, regarding the prognosis (PFS and OS) of patients with a TF <3% treated with CDK4/6i+ET, this can be found in Supplemental Fig. S7. The results show that this group of patients have an improved PFS compared to the other groups. However, we did not observe a statistically significant association with OS.

Then the authors could ask the questions:

-of 150 detectable signatures described in the tumor how many do we detect in plasma? I could not find this information explicitly provided. Could have missed it. One should be able to see this perhaps by comparing the number of rows in the heatmaps in figure 3 and 4.

We thank the reviewer for this comment. Like tissue, all signatures were identified in plasma in patients with a TF $\geq 3\%$ using the available ctDNA signal. In another comment below, we provide more data of the correlation of DNA segments and DNA signatures in paired tumor versus plasma samples (see below).

-Are some of these signatures over/underrepresented in plasma compared to tumor?

We thank the reviewer for this comment. According to the correlation coefficients of the 150 signature scores evaluated in paired tissue versus plasma samples (Supplemental Table 6), we have observed that none of the 4 immune-related DNA-based signatures are statistically significantly correlated and the correlation coefficients are very low (i.e., 0.26, 0.18, -0.02 and -0.08). Thus, detection of tumor immune-related biological processes from plasma ctDNA might be challenging. We have now added this comment in the discussion section, in "limitations": "*Seventh, ctDNA-signatures might not capture well tumor immune-related biology as tissue DNA-signatures.*"

-What is the correlation of each signature to tumor burden and TF.

Here the authors present the results unadjusted ctDNA scores (TF correlates with overall number of detected CNA alterations and with 46 of 150 (31.0%) ctDNA-based signature scores), and adjusted for TF ctDNA-based signature scores (here the correlation to TF of course disappears, this gives a possibility for a more unbiased (byTF) comparison of ctDNA signatures from patient to patient.

We thank the reviewer for this comment. The correlations of each signature with TF (unadjusted and adjusted) are reported in Supplemental Table S3. We have now added

p-values. Regarding the correlation of each signature with tumor burden, we have decided to remove it based on the comment/suggestion from Reviewer#2, since there is no objective or perfect way to 1) measure tumor burden, or 2) define tumor burden from the clinical perspective

Then it is a bit unclear, do the authors continue the analyses (from line 177 on) with the adjusted or not adjusted ctDNA scores? F.ex. in the case of the comparison of tumor burden (as number of met sites) to TF to ctDNA scores? Because the ctDNA-based signatures that highly correlated with TF represented relevant biological processes, reducing their effect in the posterior analyses with adjusted data may be unfair?

We thank the reviewer for asking for this clarification, and we apologize for the confusion. The large majority of the analyses from line 177 have been done using the signatures adjusted by TF. For clarity, we have now specified every time, in the text, whether the signatures were adjusted or unadjusted every time they are cited.

A moderate correlation between tumor and plasma is observed for around 40 ctDNA signatures, increasing to 60 if the timepoint of sample taking is closer. One could highlight some of those in the text as well say if the 60 contain the 40. It is readable from comparing the suppl tables, but still.

As suggested by the reviewer, we have now highlighted 2 of the top signatures highly correlated between tumor tissue and plasma, which are:

UNC_8q_Amplicon_Median_BMC.Med.Genomics.2011_PMID.21214954
UNC_Scorr_P53_Mut_Correlation_BMC.Cancer.2006_PMID.17150101

In addition, we have now clarified that 36 of the 40 (90%) signatures, which were found correlated in the overall "paired" population, are also contained within the 63 signatures found in paired samples obtained on the same timepoint (Supplemental Table S4). Of note, this last group has a lower number of samples (n=27) than the overall population (n=54), which should affect the power to detect significant differences.

Comments to figures:

Figure 1.

1A. Can take the space of signature 1, signature 2, signature 3...to instead illustrate all the compared pairs (tumor-plasma, before after treatment)

As suggested by the reviewer, we have now modified Fig. 2A (former Fig. 1A) to better illustrate what has been performed in our study.

1B. On the X axis add (TF) to Tumor fraction (TF) in order to signify that it is the same thing that is adjusted for in the right panel (one may get confused and think it is pathologically assessed tumor fraction in the tumor). Also a brief mention about how TF is calculated in the legend may help in addition to the Method section.

As suggested by the reviewer, we have added how TF is calculated in the figure legend. Regarding former Fig. 1B (now Fig. 2), we have now modified this entire figure to better

reflect all the analyses/results using paired plasma versus tissue samples and, as suggested by reviewer#2, we removed Fig. 1B.

1C. X axis, I would replace "tissue based" with "tumorDNA-based"

As suggested by the reviewer, we have changed "tissue-based" for "tumor DNA-based" in Fig. 2B-C (former Fig. 1).

Figure2.

D and E should be swapped, both to accommodate the left to right reading pattern, and also because the hazard ratio information follows more naturally the survival curves.

As nicely suggested by the reviewer, we have swapped both sections of Fig. 3 (former Fig. 2).

Figure D (which I suggest should become E) requires specific attention, as the columns need to be better explained. Of "Average ctDNA signal of 16 features of the RB-LOH DNA-based signature (column on the left), weight and direction of each feature (column in the middle) in the original signature as reported in Xia et al.(13) and mean change of the 16 features (column on the right) in 7 patients with paired plasma samples only the third is clear to the reader. How these values are obtained and what they mean should be explained in the legend instead of referring the reader to another paper. "original tissue based weight" cannot say much to one who would like to understand what was done just from the figure.

As suggested by the reviewer, we have now better clarified the labels of Fig. 3E (former Fig. 2D). Regarding the label "original tissue based weight", we have changed it to: "DNA segment weights", to better clarify that these are the weights of each segment within the RB-LOH signature.

Figures 3 and 4 are, in my mind, somewhat less central to the present report. Would this subclassification based on ctDNA from plasma samples from this particular set of patients and its correlation to PAM50 and inlclust be clinically relevant? What more does it say in addition to what was already estimated by the direct tumor-plasma correlation of ctDNA pathways? Maybe good to have. The data in Figure 4- was it not the main topic of the much referred to reference 10? I mention that just because a lot of attention also in the text is given to describe these results, which seem to me less central to the very important message this paper brings.

We thank the reviewer for this comment. In our opinion, Figures 4 and 5 (previously 3 and 4) bring new and relevant results worth reporting here. In Figure 4, we show, for the first time, the ability to identify tumor subtypes in plasma samples based on CNA-based data, as done with gene expression in tumor tissue. These ctDNA-based subtypes recapitulate the known RNA-based luminal versus non-luminal classification and are strongly prognostic (for both PFS and OS) in our combined series of 152 patients treated with CDK4/6i+ET. In Figure 5, we show that these tumor subtypes identified from plasma ctDNA-based data are very well recapitulated in tumor tissue DNA, and their clinical behavior in advanced disease (i.e., survival outcome) is recapitulated in early disease.

Overall, this result in tumor tissue reinforces the value of DNA-based tumor subtyping in breast cancer (whether plasma or tumor tissue is used) but, more importantly, reinforces the value of all our prior findings in plasma.

Reviewer #2 (Remarks to the Author): clinical expertise in breast cancer and CDK4/6 inhibitor response

Analysis of ctDNA is a valuable tool that has proven to be useful in identifying mutations and copy number alterations for targeted therapy. Here, the authors apply machine learning model, previously developed using tissue DNA, to detect 150 multi-gene signatures in ctDNA with the aim of identifying biologic features of the cancer including measures of estrogen receptor signaling and tumor proliferation. They examined a data set of 246 plasma samples, mostly from patients with advanced HR+/HER2- breast cancer, but included only those that had a tumor cell fraction $\geq 3\%$ (178 samples). Using a subgroup of 54 patients with paired ctDNA and tumor DNA, they found very limited correlation (0.4 average) between tumor and plasma signatures, raising the questions of what is really being measured by the ctDNA signatures if it is not reflective of the tumor tissue signatures, which is not addressed by the authors.

We thank the reviewer for raising this concern. We agree that the correlation coefficients are not 0.9-1.0. However, we must consider that the biology of a tumor tissue biopsy is expected to differ from the biology of a plasma sample. For example, the samples were not obtained on the same date. In addition, plasma samples are likely to recapitulate an "average" biological state from many metastatic lesions, whereas tumor biopsies reflect the biological state of one tumor lesion. Nonetheless, the correlation coefficients are significant, and improve if the time gap between the date where the samples were obtained is close. Moreover, the plasma-based RB-LOH signature better predicts PFS and OS than the same signature evaluated in tumor tissue (current Fig. 3D).

To further demonstrate that CNA-based data from plasma allows the identification of similar biological states as in tissue, we have estimated the intra-patient correlation coefficients across the CNA-based signals of 514 DNA segments using 54 paired samples (tumor tissue versus plasma). Overall, 57% of patients had a correlation coefficient between plasma and tissue >0.50 , which increased to 83% when we evaluated 29 patients with plasma TF $>10\%$. In these 29 patients with a plasma TF $>10\%$, 59% and 24% patients had a correlation coefficient of >0.70 and >0.80 , respectively (table below). We have now added these results in Supplemental Material.

Overall, these results strongly suggest that plasma ctDNA can reliably capture CNA-based signals from tumor tissue, although the amount of ctDNA might impact the ability to accomplish this. This limitation has now been acknowledged in the discussion section, limitations paragraph: *"First, ~39% of patients had a TF $<3\%$, and ctDNA-signatures could not be evaluated. In addition, ~30% of patients had a TF of 3-10% and this might limit the detection of the ctDNA-based signatures. Further studies evaluating deeper ctDNA sequencing strategies and signature detection in patients with very low TF, including those with early-stage disease, is warranted. For example, expressed genes might be inferred by evaluating nucleosome footprints from whole-genome sequencing of plasma DNA(37)."*

Table S5. Proportion of patients with a certain correlation coefficient when 514 DNA signals are compared between paired plasma and tissue.

	TF >10%	TF 3-10%	ALL
	n=29	n=25	n=54
Cor >0.80	24%	4%	15%
Cor >0.70	59%	12%	37%
Cor >0.60	62%	20%	43%
Cor >0.50	83%	28%	57%
Cor >0.40	90%	36%	65%
Cor >0.30	97%	44%	72%
Cor >0.20	97%	48%	74%
Cor >0.10	97%	64%	81%
Cor <0.10	3%	36%	19%

Figure. Correlation between plasma and tissue of 514 DNA signals in a single patient. Correlation coefficient = 0.928.

Finally, we evaluated if the ctDNA-based RB-LOH is significantly associated with PFS and OS in 71 patients with a TF 3-10% before starting endocrine therapy in combination with a CDK4/6 inhibitor. Like the overall population, we observed a statistically significant association with both clinical endpoints (PFS: HR=1.32, p=0.023; OS: HR=1.54, p=0.011). This results strongly suggests that our approach can work, even in patients with a TF 3-10%. This new result can now be found in Supplemental Material.

The authors provided evidence supporting correlations between ctDNA proliferation signatures and tumor cell fraction, luminal A signatures and bone-only disease, luminal signatures and ER+ status, and HER2 signatures with HER2+ disease. Additionally, using a set of 87 pre-treatment plasma samples, they identified ctDNA signatures associated with poor prognosis and response to treatment with endocrine therapy and CDK4/6 inhibition, and focused on RB-LOH signature. Using an independent cohort, which after filtering to tumor fraction >3% was only 65 patients, they found that RB-LOH signature in ctDNA was associated with worse PFS. They also examined a cohort from MSK and found similar association between RB-LOH in tumor tissue taken <1 year from starting therapy and PFS – note that this was signatures derived from tumor tissue and ctDNA signature was not reported for the MSK set.

Additionally they performed unsupervised clustering on ctDNA signatures and found 4 clusters that were associated with PFS and OS in both ctDNA validation cohort and METABRIC tumor tissue cohort. Overall the authors conclude that applying their machine learning model to identify 150 signatures in ctDNA provides additional information about disease biology.

Major Comments:

- 1) The abstract needs to be restructured to reflect the methods and data presented in the paper, it is now very general and vague.**

As suggested by the reviewer, we have improved the 150-words abstract to better reflect the methods and data presented in the article: *"Liquid biopsy has proven valuable in identifying individual genetic alterations; however, the ability of plasma ctDNA to capture complex tumor phenotypes with clinical value is unknown. To address this question, we performed 0.5X shallow whole-genome sequencing in plasma from 459 patients with metastatic breast cancer, including 245 patients treated with endocrine therapy and a CDK4/6 inhibitor (ET+CDK4/6i) from 2 independent cohorts. We demonstrated that machine learning multi-gene signatures, obtained from ctDNA, identify complex biological features, including measures of tumor proliferation and estrogen receptor signaling, similar to what is accomplished using direct tumor tissue DNA or RNA profiling. More importantly, a ctDNA-based genomic signature tracking retinoblastoma loss-of-heterozygosity, and newly discovered DNA-based subtypes, were found significantly associated with poor response and survival outcome following ET+CDK4/6i, independently of plasma tumor fraction. Our approach opens new opportunities for the discovery of additional multi-feature genomic predictors coming from ctDNA in breast cancer and other cancer-types."*

- 2) The authors note initial signatures were developed as per Reference #10. It is the understanding of this reviewer that the initial signatures were developed from TCGA data, which include early stage tumors. The current paper focuses on advanced breast tumors. The signatures in advanced tumors may differ from those seen in early disease. One example is ESR1 mutations, rarely seen in early tumors, but commonly emerge in patients treated with endocrine therapy.**

We thank the reviewer for this comment. Indeed, metastatic disease acquires genetic alterations (such as ESR1 mutations) compared to early-stage disease. At the RNA expression level, we and others have shown that there is a higher proportion of non-luminal subtypes (i.e., HER2-enriched and Basal-like) in HR+/HER2- disease in the metastatic setting compared to the early-stage setting (Prat et al. JAMA Oncol 2016; Prat et al. JCO 2021, Cejalvo et al Cancer Res 2017; Aftimos et al. Cancer Discovery 2021). In many instances, there is a true shift of tumor subtype from luminal A or B in primary disease to non-luminal in the metastatic setting. Importantly, phenotypic profiling using intrinsic subtyping (i.e., PAM50) of metastatic tumors provides strong prognostic value (Brasó-Maristany et al. Mol Oncol 2021; Prat et al. JCO 2021). Of note, the PAM50 assay was developed in primary tumors, and still "works" as a prognostic/predictive tool in the advanced setting, as demonstrated in MONALEESA-02/03/07 trials (Prat et al. JCO 2021), EGF30008 trial (Prat et al. JAMA Oncol 2016) and BOLERO-2 trial (Prat et al. Oncologist 2019).

To address this comment, we analyzed the 150 DNA-based signatures across 158 tumor tissues (either primary tumors or metastatic tumors). As shown in the heatmap below, the identification of the 4 clusters/subtypes is independent of the tissue profiled (i.e., primary versus metastasis). However, the frequencies of Clusters 1-4 differ. For example, the proportion of primary tumors falling in Clusters 1, 2, 3 and 4 was 47.6%, 27.0%, 11.1% and 14.3%, respectively. The proportion of metastatic tumors falling in Clusters 1, 2, 3 and 4 was 27.0%, 39.3%, 19.1% and 14.6%, respectively. Thus, an enrichment for cluster 1 (better prognosis) was observed in primary tumors compared to metastatic tumors (p=0.0103). This enrichment for aggressive tumor biology in metastatic disease is consistent with the results of previous studies using gene expression-based data (cited above). Overall, we conclude that, although individual genetic differences exist between primary and metastatic disease, the biological processes captured by our signatures can be identified in both clinical settings (i.e., metastatic, and early), despite the signatures being derived originally from early-stage tumors. We have now included this result in Supplemental Material.

Figure. Proportion of cases in each cluster based on tissue-type (primary versus metastatic). Cluster 1: 47.6% in primary vs. 27.0% in metastatic (p-value=0.0103)

Figure. Unsupervised hierarchical clustering of 150 DNA-based signatures across 158 tumor tissue samples (n=63 primary tumors and 89 metastatic tumors).

2) The paper is very dense and contains vast amount of data including many supplementary files. The authors should state their specific hypotheses and consider organizing the results and methods accordingly. They may also consider splitting the paper into two separate ones.

We acknowledge this is a dense study with lots of data and results. However, we would like to keep it as one single article, which will be the basis of future work by our group and others on this area. As suggested, we have now clarified better the specific hypotheses in the introduction section: *"Here we hypothesized that DNA-based signatures tracking breast cancer biological processes can be detected in ctDNA and provide clinically useful information. In addition, we hypothesized that the 150 DNA-signatures in plasma and tissue can help identify tumor subtypes within hormone receptor-positive and HER2-negative breast cancer (HR+/HER2-)."*

3) The descriptions of the cohorts studied in this paper are not clear. Consider assigning a number or letter to each cohort and use these consistently throughout the manuscript.

We agree that more clarity is needed. As suggested by reviewer#1 as well, we have now added a new Figure 1 (see below) that identifies and describes each cohort. In addition, we have now labeled each plasma cohort as Plasma-1, Plasma-2, CDK-Validation-1 and CDK-Validation-2 cohorts. These cohort IDs are used throughout the manuscript. In addition, we have now moved the Supplemental Table with the clinical-pathological characteristics of CDK-Validation-2 cohort to the main Table 2, where the clinical-

pathological features of CDK-Validation-1 cohort were shown. This will allow a better understanding of the different cohorts used throughout the study.

New Figure 1.

Figure 1

4) Consider adding a table describing each of the cohorts, including prior treatment, and response to treatment. The knowledge of prior treatment is important as the dynamics of the markers may change over time.

Although it would be valuable information, the heterogeneity of the prior treatments received and the variability in the duration of these treatments makes it difficult to draw conclusions. Nonetheless, we have been able to retrieve "type of prior therapy" (i.e., none, endocrine therapy, chemotherapy, or both) before initiating endocrine therapy in combination with a CDK4/6 inhibitor for the first validation CDK cohort (CDK-Validation-1). This information has now been added in Table 2, which now also includes the clinical-pathological data of the CDK-Validation-2 cohort.

5) It appears that sample collection was prospective, but additional patient, tumor, and treatment information was collected from the patients' medical records. Please discuss possible bias.

The two independent datasets which were used to link the genomic data with clinical outcome do not come from clinical trials and thus are subjected to potential inherent biases such as patient selection or subjectivity in determining drug response or progressive disease. We have now described these potential biases in the discussion section, study limitations paragraph: "Fifth, although both independent datasets collected plasma samples prospectively from patients treated with endocrine therapy and a CDK4/6 inhibitor, the cohorts are not from clinical trials and are prone to potential biases such as patient selection, inconsistent evaluation of the disease during therapy and subjectivity in determining drug response and progressive events".

6) Sample size considerations and power for each analysis are not included. Are the cohorts simply samples of convenience? Further explanations are required to understand the true level of evidence of each of the reports.

The objective of this study was to provide, for the first time, proof that DNA-based signatures tracking breast cancer biological processes can be detected in ctDNA from metastatic breast cancer and could have clinical implications. In our view, this is an exploratory study that shows clinical validity at this point. Further studies are needed to determine Level 1 evidence and, more importantly, clinical utility. We did not perform a sample size calculation and we used all the samples available for correlative analyses. This has been added as a limitation in the discussion section: "*Second, this was an exploratory study, and no formal sample size calculation was performed. Thus, the lack of a formal design through a pre-planned analysis prohibits inference of negative results*".

7) Please provide p-values for the correlations between tissue-based signatures and ctDNA-based signatures (page 6, starting line 195)? Specifically:

As suggested, we have now added p-values to Supplemental Table S6.

a. The correlation coefficients, even for samples obtained within 8 weeks, are pretty low, but hard to interpret as no p-values given except in the 2 examples in Figure 1C. If there is no, or only limited, correlation between the tissue signature and the ctDNA signature, then what exactly is being measured in the ctDNA? These data should be presented first, and needs to be explained, because all the other correlations reported could be irrelevant if the signatures detected in ctDNA are not reflective of signatures obtained from tumor tissue.

As suggested, we now show the p-values in Supplemental Table S6. Across the 54 patients with paired tumor versus plasma samples, 78% of the DNA-based signatures (117/150) show a statistically significant result. When we evaluate the signals from the 514 DNA segments, 450 (87%) are found statistically significantly correlated with a median correlation coefficient of 0.47. In addition, 93 of 514 (18%) segments have a correlation coefficient >0.60. Correlations from signals from DNA segments have now been added in Supplemental Material.

We agree with the reviewer that the correlation coefficients are not perfect (i.e., 0.9-1.0). However, we must consider that the biology obtained from a tumor tissue biopsy is expected to differ from the biology obtained from a plasma sample. For example, the samples were not obtained on the same date. In addition, plasma samples are likely to recapitulate an "average" biological state from metastatic lesions, whereas tumor biopsies reflect the biological state of just one lesion. Nonetheless, the correlation coefficients are significant for the majority of signatures and DNA segments, and the correlation coefficients increase if the time gap between both samples is closer. Moreover, we show that the plasma-based RB-LOH signature better predicts PFS and OS than the same signature evaluated in tumor tissue (Fig. 3D).

Moreover, we evaluated if the ctDNA-based RB-LOH is significantly associated with PFS and OS in 71 patients with a TF 3-10% before starting endocrine therapy in combination

with a CDK4/6 inhibitor. Like the overall population, we observed a statistically significant association with both clinical endpoints (PFS: HR=1.32, p=0.023; OS: HR=1.54, p=0.011). This results strongly suggests that our approach can work even in patients with a TF 3-10%. This new result can now be found in Supplemental Material.

To further demonstrate that CNA-based data from plasma allows the identification of biological states in tissue, we calculated the intra-patient correlation coefficients of the CNA-based signals across 514 DNA segments using 54 paired samples (tumor tissue versus plasma). Overall, 57% of patients had a correlation coefficient >0.50, which increased to 83% when 29 patients with a plasma TF >10% were evaluated. In these 29 patients with a plasma TF>10%, 59% and 24% had a correlation coefficient of >0.70 and >0.80, respectively (table below). Overall, these results strongly suggest that plasma ctDNA can reliably capture CNA-based signals from tumor tissue, although the amount of ctDNA might impact the ability to accomplish this. This limitation has now been acknowledged better in the discussion section, limitations paragraph: *"First, ~39% of patients had a TF <3%, and ctDNA-signatures could not be evaluated. In addition, ~30% of patients had a TF of 3-10% and this might limit the detection of the ctDNA-based signatures. Further studies evaluating deeper ctDNA sequencing strategies and signature detection in patients with very low TF, including those with early-stage disease, is warranted. For example, expressed genes might be inferred by evaluating nucleosome footprints from whole-genome sequencing of plasma DNA(37)."*

Table S5. Proportion of patients with a certain correlation coefficient when 514 DNA signals were compared between paired plasma and tissue.

	TF >10%	TF 3-10%	ALL
	n=29	n=25	n=54
Cor >0.80	24%	4%	15%
Cor >0.70	59%	12%	37%
Cor >0.60	62%	20%	43%
Cor >0.50	83%	28%	57%
Cor >0.40	90%	36%	65%
Cor >0.30	97%	44%	72%
Cor >0.20	97%	48%	74%
Cor >0.10	97%	64%	81%
Cor <0.10	3%	36%	19%

Figure 2C. Correlation between plasma and tissue of 514 DNA signals in a single patient. Correlation coefficient = 0.928.

b. Later in the paper, the authors highlight the correlation between 48 features of tissue based RB-LOH signature and ctDNA, suggesting that correlation is between tissue and ctDNA is important (page 8).

We thank the reviewer for this comment, which is in line to our previous response. The fact that the main 48 features of the RB-LOH signature, which was derived from primary tumors from the TCGA dataset, are well captured in plasma ctDNA in advanced disease, and in the "right direction" (i.e., features with a negative weight have a negative signal in ctDNA and features that have a positive weight have a positive signal in ctDNA), argues in favor of plasma ctDNA as a reliable tool to measure the status of this signature. This hypothesis is further confirmed by the fact that samples at progression to endocrine therapy and CDK4/6 inhibition show an increase in the RB-LOH score in plasma compared to baseline samples before starting therapy, and this is due to the "right" biological changes in the signal of each DNA segment of the RB-LOH signature, such as the decrease in the 13q14 segment where RB1 gene is located.

c. On page 13, starting on line 390, the authors describe the correlation of 6 PAM50 RNA-based tissue signatures with each of the 150 ctDNA-based signatures, as well as correlation with gene expression in the n Counter Breast Cancer 360 Panel. This needs to be moved up in the paper and should be discussed along with relationship between ctDNA and tissue DNA signatures.

As suggested, we have now moved this result up in the article in the new Figure 2 (previously Figure 1), where we describe the correlation of ctDNA-based signature with DNA/RNA-based tissue data.

d. What is the association between tissue based signatures and the other features presented (i.e. bone only disease, tumor burden, ER status, HER2 status, response to endocrine therapy, etc)? Are tissue based signatures or ctDNA based signatures better associated with these features? It is mentioned on line 304 that RB-LOH ctDNA signature was associated with PFS but not RB-LOH tumor signature, suggesting that ctDNA based signature may be better, but what about the other signatures?

As suggested by the reviewer, we have now compared the ability of the tissue-based signatures versus the ctDNA-based signatures to better predict PFS and OS in 28 patients with paired data in the CDK-VALIDATION-1 cohort. The result is now in Supplemental Material. Of the 150 DNA-based signatures, 17% and 13% were statistically significantly associated with PFS and OS, respectively, when evaluated in plasma. When the same signatures were evaluated in tissue, only 2% and 1% were statistically significantly associated with PFS and OS, respectively. Thus, ctDNA-based signatures are better in predicting survival outcomes in advanced breast cancer treated with endocrine therapy and CDK4/6 inhibition than the same signatures when evaluated in tissue. As argued in the manuscript, the potential explanation is that plasma-based signatures capture "the most up to date" biological state of the disease before starting therapy.

d. The authors comment in the introduction "the type of metastatic organ or site can compromise the expression patterns obtained from bulk RNA and might not reflect the intra-patient tumor heterogeneity" (p4, line 125) – perhaps this is the cause of the lack of correlation with tissue? It would be helpful if authors included the location of tissue biopsy specimen for metastatic patients and assessed correlation by metastatic site. What is the impact of the different sequencing methods used (shallow WGS for ctDNA vs targeted panel sequencing for tissue)? How dependent on

sequencing method is the machine learning model? The overall lack of correlation between tissue and ctDNA signatures needs to be investigated further/better explained before anything else in the paper can be interpreted.

Our comment in the introduction about RNA expression profiling of metastatic tissue is based on our prior work reported by Brasó-Maristany et al. Mol Oncol 2020 (PMID: 34051058). There, we observed that metastatic organ site might affect the expression of some genes; in particular, 74 of 771 genes analyzed were organ-specific and subtype independent. Some of these genes, like keratin-5 (KRT5) expression in skin might impact the tumor subtype classification of Basal-like disease, for example. Nonetheless, RNA-based expression in metastatic breast cancer is reliable and can provide prognostic information, as shown by us and other groups.

Regarding evaluating the organ site of the metastatic biopsy performed for tissue-based DNA-seq analysis, unfortunately we do not have this information and it would take 4-6 months to obtain. The type of metastatic organ biopsied could indeed impact the correlation of signatures with plasma ctDNA-based data. However, we feel it is beyond the scope of this study to evaluate this thoroughly, since this would require a very large sample size of tissue biopsies from many different organs, and the focus of our study is plasma ctDNA and not tissue DNA. Of note, the MSKCC tissue-based DNaseq metastatic dataset used in our study shows that the prognostic value of the RB-LOH tissue-based signature is strong, despite different types of tissue being used (primary or any metastatic site). Thus, this argues in favor of our tissue DNA-based signatures to reliably detect the correct tumor phenotype and be somewhat less affected by the type of metastatic organ.

Regarding the type of sequencing methods and how they might affect the detection of the DNA signatures, the results from Xia et al. (Nat Commun 2019) and our study show that the 150 signatures can be detected successfully across a variety of sequencing approaches: TCGA (whole exome sequencing), METABRIC (Affymetrix SNP 6.0), MSKCC cohort (an in-house targeted sequence panel of ~400 genes), shallow WGS plasma ctDNA and the VHIO tissue panel (a custom hybridization-based capture panel targeting 435 genes with an in-built genome-wide SNP backbone targeting 20000 SNPs). The most plausible explanation is that we are measuring the signals from large chromosomal regions (55% of the segments have >1.5 million bases, and 71% of the segments have >0.8 million bases and 89% of the segments have >0.2 million bases – see response to reviewer#3 regarding this topic). In addition, the ~500 DNA segments were selected in Xia et al. because they represent highly recurrent events in breast cancer. Said that, more fine tuning of the methodology used for signature detection in tissue and plasma is needed before implementing these biomarkers in the clinic. This is currently work ongoing by our group to analytically validate and standardize the tissue and plasma DNA signatures.

Finally, we would like to point out that there is a significant correlation between tissue and plasma signatures, as discussed above. In this direction, another aspect to highlight is that plasma ctDNA was used to derive the 4 cluster/subtype-based predictor. This predictor was applied to tissue-based DNA data such as METABRIC (Affymetrix SNP 6.0) and MSKCC cohort (an in-house targeted sequence panel of ~400 genes), and these subtypes showed the expected prognostic association as in our 2 plasma validation

cohorts. Overall, this data, together with the results discussed previously, strongly suggest that ctDNA-based data can reliably identify DNA-based signatures with clinical value.

8) The authors state that “The observed correlation between proliferation-related ctDNA-based biology with TF suggested that TF reflects the biological aggressiveness of the disease beyond the patient’s tumor burden” (page 6, line 177). This statement is not supported by the evidence provided. No evidence is given regarding the biological aggressiveness, such as rate of tumor growth, or rapidity of clinical progression. There is no support given for decision to use <3 versus ≥3 metastatic sites as a measure of tumor burden and I do not believe this is an accurate measure of tumor burden. Furthermore, method used to count metastatic sites was not provided (are 2 spots in the liver 2 sites or 1? How large does a spot need to be to be counted? – this needs to be included in the methods section). Imaging based estimates of volume of disease would be more meaningful if available. Otherwise, I would not refer to this as “tumor burden” in this paragraph, but rather as number of metastatic sites. The lack of correlation between proliferation-related signatures and “tumor burden” could in fact reflect conclusions other than those given by the author including 1) proliferation signatures are not good markers of tumor burden, or 2) tumor burden is not well measured by greater or less than 3 metastatic sites. The correlation presented between luminal A signatures and bone only disease is more compelling. If the authors are not able to better support their claims regarding tumor burden, then the findings related to tumor burden should be omitted or else very carefully stated with much more limited conclusions.

We agree with the reviewer that “tumor burden” is very difficult to quantify or even define objectively. We chose the definition of “less than 3 metastatic sites” or “3 or more metastatic sites” since it has been used in clinical trials in metastatic breast cancer to define high vs low tumor burden (for example, in the MONALEESA-02 phase III trial ([link: https://bit.ly/3TaAM1g](https://bit.ly/3TaAM1g))). Nonetheless, we agree we cannot conclude “the lack of association of TF with tumor burden” with just 1 analysis. Therefore, we have decided to remove the entire paragraph from the manuscript, including Supplemental Table 4 and 5.

9) Why did the authors focus on the RB-LOH signature? This was not the signature with the lowest p-value or highest HR. Without an explanation regarding why this signature was picked to focus on, it seems like cherry-picking. There is some discussion of relevance of RB-LOH in the discussion section, some of this should be included in the results to justify why the authors focused on this.

As suggested, we have now included a sentence in the results section to justify why we choose to further explore this signature: “A high score of a ctDNA-signature tracking RB-LOH(19) was found one of the top biomarkers associated with poor outcome and treatment response (Fig. 2B-C). Since loss of RB is a known mechanism of resistance to CDK4/6 inhibitors(22,29,31,32), we decided to focus on this signature, which is composed of 224 copy number features, including amplification of 2p (e.g., ETV6), 3q (e.g., PIK3CA), 8q (e.g., MYC), 20q (e.g., AURKA) and 21q (e.g., Tmprss2 and ERG), and deletion of 2q (e.g., PARD3B), 4q, 5q, 12q, 13q (e.g., RB1), 15q and 17p.”

10) What is the association between ctDNA-based tumor subtypes (clusters 1-4, page 12) and clinical factors such as prior treatment, response to treatment, location of metastatic disease, etc? What, if any, is the clinical relevance of identifying these clusters? I found this discussed on page 14, beginning at line 417. This should be reorganized so the discussion of the clusters goes with the clinical meaning (PFS, OS) of the clusters.

As suggested, we have now better justified in the results section why we aimed to identify subtypes in ctDNA: "*Phenotype-based classification in metastatic breast cancer using tumor tissue RNA expression profiling such as intrinsic subtyping (i.e., Luminal A, Luminal B, HER2-enriched and Basal-like) is prognostic and might predict treatment benefit(5-7). To evaluate if the biology displayed by the 150 ctDNA-based signatures can identify subtypes with clinical relevance, we performed an unsupervised hierarchical cluster analysis of all 150 signatures across 178 plasma samples with a TF \geq 3% (Fig. 3A).*" Regarding the relationship between clusters 1-4 and treatment response, we observed an overall response rate (ORR) of 52.73%, 34%, 7.14% and 16.67% in Cluster 1, Cluster 2, Cluster 3 and Cluster 4, respectively ($p < 0.001$). Regarding the relationship between cluster 1-4 and prior endocrine sensitivity, this was observed in 91.67%, 80.0%, 57.14%, and 55.56% in Cluster 1, Cluster 2, Cluster 3 and Cluster 4, respectively ($p = 0.004$). We have now included the ORR and prior endocrine sensitivity results in Figure 4C, all of which support the differences in biology observed in these 4 subtypes. Finally, we do not have information regarding prior treatments, so we cannot provide this data.

Figure 4C. Association of ctDNA-based clusters with response and prior endocrine sensitivity in patients treated with endocrine therapy and CDK4/6 inhibition.

11) Given some of the comments listed above, it appears that the results are still hypothesis generating and the conclusions should be revised accordingly.

In our opinion, our study proves that CNA-based data can provide clinically relevant information in metastatic breast cancer, specifically in HR+/HER2- disease treated with endocrine therapy and CDK4/6 inhibition. Our future work is to standardize the assay and demonstrate its clinical utility.

Minor Comments:

1) The authors characterize solid tumor biopsy in the metastatic setting as “challenging.” Depending on location, biopsy of metastases can actually be relatively straight forward. I agree that liquid biopsy has multiple advantages over solid tissue biopsy, including less invasiveness to the patient. I advise the authors to adjust their language around solid tumor biopsy to be more specific regarding the perceived disadvantages vs advantages of liquid biopsy.

We agree with the reviewer. We have now stated that only “sometimes” solid tumor biopsies are challenging: “...tumor invasive biopsy procedure, which can be quite challenging sometimes in the metastatic setting.”

2) In line 546, the author state “To date, DNA sequencing has identified few FDA-approved actionable genetic alterations in cancer, especially breast cancers” and then go on to discuss that RNA expression phenotypes are associated with treatment benefit, implying that that expression signatures are more actionable. This is not supported by the literature. This sentence should be omitted or re-written

We have re-phrased the sentence: “To date, DNA sequencing has identified few FDA-approved actionable genetic alterations in cancer, especially breast cancers. Phenotypic characterization using multi-gene RNA-based expression might add new biological and clinically relevant information. However, implementation of tumor-based RNA-based gene expression profiling in the metastatic setting is a major challenge since tumor tissue is often not readily available.”

3) Figure 4 legend suggests that unsupervised cluster analysis was performed on the METABRIC tumor sample set. My reading of the paper was that cluster analysis was performed on the ctDNA data set, fixed, and then applied to the METABRIC dataset. This should be clarified.

Figure 5 (former Figure 4) represents an unsupervised cluster analysis on METABRIC tumor sample set. The identification of cluster 1-4 in each sample, which can be seen below the data matrix, is based on the ctDNA dataset, which was fixed, used to derive the predictor, and then applied to the METABRIC dataset. We have now better clarified this in Figure 5 legend: “Figure 5. DNA-based tumor profiles in tissue samples and association with clinical outcomes. (A) Unsupervised cluster analysis of 1,689 tumor samples (columns) from METABRIC dataset(26) and the 150 DNA-based signatures scores (rows). Orange and violet colors represent scores above and below the median value of the signature across the dataset. Below the array tree, the InctClust classification(26) and the PAM50 molecular subtypes are shown for each sample. The 4 clusters/subtypes identified using a ctDNA-based subtype predictor are shown below the data matrix”.

Reviewer #3 (Remarks to the Author): expertise in ctDNA profiling in breast cancer

Prat and colleagues propose to use ctDNA profiling for (noninvasively) measuring several biological features in the metastatic breast cancer. Building this upon their previous work, they use copy number alterations to estimate gene expression activity of 150 signatures. Due to low levels of ctDNA in early-stage breast cancer, the focus of the paper is on metastatic setting, where in most cases, (tumor) allele fractions are expected to be at least in single digits.

Cell-free DNA profiling is done by shallow whole genome sequencing (~0.5X). The idea of using copy number patterns to capture tumor phenotypes is very interesting and could be valuable in metastatic settings. This would be particularly useful due to its very low cost.

Major comments:

- 1- The authors use ichorCNA to evaluate tumor fractions. Unfortunately, while ichorCNA has been used in several studies so far, its performance is not well established (for both TF estimation & CNV detection); and therefore, before applying to the data generated in this study, one would need to know its sensitivity for CNV detection as a function of ctDNA conc. & sequencing coverage. Particularly, I would like to see whether ichor estimates (1) correlate well with SNV-based TFs, and (2) are sufficiently close to zero for non-cancer controls.**

We thank the reviewer for this comment. In Adalsteinsson et al. (Nat Communic 2017), a TF cutoff of 3% by ichorCNA achieved a sensitivity of 0.95 for detecting presence of tumor and a specificity of 0.91 for correctly classifying a healthy donor. To further explore if the ichorCNA TF estimates are correct, we have analyzed 26 plasma samples with shallowWGS and the standardized Guardant B360 panel (Martínez-Sáez et al, npj Breast Cancer, 2021). As shown in the table below, a correlation coefficient of 0.860 was observed between both methods.

y-axis: TF by ichorCNA
x-axis: Highest VAF by Guardant

In addition, we performed shallow WGS in 14 healthy individuals, and the log2 values by ichorCNA estimates were determined to be on average 0. In detail, below:

Table S14. Statistics of bin log2 values output from iChorCNA of 14 healthy control samples (1Mb bins)

	log2 1Mb bin sizes										
	Min.	1st Qu.	Median	Mean	3rd Qu.	Max.	bins	sd	Margin Error	CI95.lower.limit	CI95.Upper.limit
Healthy control 1	-0.100	-0.019	0.001	0.001	0.022	0.136	2510	0.032	0.001	0.000	0.003
Healthy control 2	-0.135	-0.015	0.000	0.000	0.017	0.098	2510	0.025	0.001	0.000	0.001
Healthy control 3	-0.121	-0.020	0.000	0.001	0.023	0.144	2510	0.035	0.001	-0.001	0.002
Healthy control 4	-0.115	-0.015	0.001	0.001	0.020	0.095	2510	0.028	0.001	0.000	0.003
Healthy control 5	-0.100	-0.014	0.001	0.001	0.017	0.165	2510	0.026	0.001	0.000	0.002
Healthy control 6	-0.222	-0.017	0.000	0.001	0.017	0.407	2510	0.031	0.001	0.000	0.002
Healthy control 7	-0.105	-0.017	0.002	0.002	0.020	0.171	2510	0.030	0.001	0.001	0.003
Healthy control 8	-0.099	-0.016	0.001	0.002	0.020	0.288	2510	0.030	0.001	0.000	0.003
Healthy control 9	-0.178	-0.021	0.001	0.001	0.024	0.139	2510	0.036	0.001	0.000	0.002
Healthy control 10	-0.127	-0.018	0.002	0.001	0.020	0.143	2510	0.031	0.001	0.000	0.003
Healthy control 11	-0.149	-0.021	0.001	0.002	0.026	0.124	2510	0.036	0.001	0.000	0.003
Healthy control 12	-0.148	-0.015	0.001	0.001	0.018	0.099	2510	0.026	0.001	0.000	0.002
Healthy control 13	-0.146	-0.021	0.000	0.000	0.022	0.131	2510	0.034	0.001	-0.001	0.002
Healthy control 14	-0.098	-0.018	0.000	0.000	0.019	0.093	2510	0.029	0.001	-0.001	0.001
AVERAGE	-0.131	-0.018	0.001	0.001	0.020	0.159	2510	0.030	0.001	0.000	0.002

Regarding the association with sequencing coverage, we address this point in another response to this reviewer below. Overall, the performance of ichorCNA to estimate TF seems appropriate for our study. We have now included this plot in Supplemental data.

2- Can authors comment on the TFs in this study? They seem too high (about 25-30% higher than 20%). Is that comparable with other studies? A comparison with existing data would be valuable.

As suggested, we have evaluated the recent studies by Reichert et al. Annal Oncol 2022 (<https://doi.org/10.1016/j.annonc.2022.09.163>) and Husain et al. JCO Precision Oncol 2022 (<https://ascopubs.org/doi/full/10.1200/PO.22.00261>). Both studies focused on TF in metastatic cancer across several cancer-types. In Reichert et al. (n=402 metastatic breast cancer samples), the median TF was 4% (IQR: 1-21%), meaning that 25% of patients had a TF above 21%. Overall, 164 patients of 402 (42%) had a TF above 10%. In Husain et al. (n=3,265), the median TF was 2%, and 28% had a TF above 10%. Overall, these results are in line with ours (n=603 plasmas), where the median TF across all plasma samples was 6.3% (IQR: 0-13%) and 31.2% of patients had a TF above 10%. We have now cited both articles and added this information in the manuscript.

3- It there any benefit in increasing sequencing depth? I am a bit skeptical that this method would work with 0.5X coverage (i.e., ultra low pass WGS), for ctDNA fractions less than 20%, especially since individual copy number (segment) states are fed to (pre-trained) models to calculate the signatures. Some analytical LOD analyses are needed, e.g., sensitivity of each signature vs coverage/TF.

As suggested by the reviewer, we have performed analytical analyses to validate our approach.

First, we would like to highlight that despite being considered ultra-low pass WGS by definition (0.1-0.5X coverage), our coverage is at the higher end (i.e., 0.5X). We performed

an *in-silico* downsampling of sample coverage for 12 samples in order to assess performance at different coverages. Different TFs are represented within these 12 samples. For each, a subset of aligned reads was selected to account for: 2X (40 M reads, 151 nt long), 1X (20M reads, 151 nt long), 0.5X (10M reads, 151 nt long) and 0.1X (2M reads, 151 nt long) coverages. Reported TFs were consistent across coverages, as shown in the table below.

Table S12. Tumor fraction reported by iChorCNA using shWGS at different coverage depths (2X, 1X, 0.5X, 0.1X)

Sample	Tumor Fraction - 40M reads_2X coverage	Tumor Fraction - 20M reads_1X coverage	Tumor Fraction - 10M reads_0.5X coverage	Tumor Fraction - 2M reads_0.1X coverage
CASE 1	0.0000	0.0000	0.0000	0.0203
CASE 2	0.0779	0.0803	0.0843	0.1049
CASE 3	0.0909	0.1124	0.0978	0.0990
CASE 4	0.2160	0.2180	0.2209	0.2582
CASE 5	0.3587	0.3591	0.3534	0.3495
CASE 6	0.4411	0.4405	0.4389	0.4428
CASE 7	0.4667	0.4518	0.4506	0.4431
CASE 8	0.5513	0.5497	0.5481	0.5389
CASE 9	0.5661	0.5672	0.5650	0.5812
CASE 10	0.6456	0.6448	0.6448	0.6474
CASE 11	0.6489	0.6473	0.6464	0.6629
CASE 12	0.6548	0.6430	0.6493	0.6291

The bin-to-bin (1Mb) log₂ values of these samples correlated as follows (correlation is shown between 2X and each of the remaining conditions) (see table below). There is a decrease in correlation of log₂ values reported by iChorCNA for the 0.1X condition (although average correlation is 0.828) in samples of TF<10%, although 0.5X (the coverage used in the presented study) performs similarly to 1X.

Table S13. Correlation of bin-to-bin log₂ values reported by iChorCNA using shWGS at different coverage depths (2X, 1X, 0.5X, 0.1X).

Sample ID	bin-to-bin log ₂ at 2X*	bin-to-bin log ₂ at 1X*	bin-to-bin log ₂ at 0,5X*	bin-to-bin log ₂ at 0,1X*
CASE 1	1	0.7591	0.6939	0.3978
CASE 2	1	0.9946	0.9909	0.9651
CASE 3	1	0.8895	0.8293	0.5832
CASE 4	1	0.9712	0.9548	0.8389
CASE 5	1	0.9764	0.9621	0.8652
CASE 6	1	0.9969	0.995	0.9794
CASE 7	1	0.9868	0.9784	0.9175
CASE 8	1	0.997	0.9952	0.9794
CASE 9	1	0.9942	0.9906	0.9615
CASE 10	1	0.9972	0.9954	0.9807
CASE 11	1	0.8382	0.7586	0.5064
CASE 12	1	0.9945	0.9904	0.9621
AVERAGE	1	0.9496	0.9279	0.8281

*all comparisons are referred to 2X coverage

Next, we applied our CNA signatures on the 11 samples with TF>3%. Again, 0.5X coverages raise similar profiles across all signatures to 1X and 2x.

Table S14. Correlation of 150 CNA signatures run on samples that have been in-silico diluted to different coverage depths

Coverage	CASE 2	CASE 3	CASE 4	CASE 5	CASE 6	CASE 7	CASE 8	CASE 9	CASE 10	CASE 11	CASE 12	AVERAGE
40M	1	1	1	1	1	1	1	1	1	1	1	1
20M	0.85	0.97	0.99	1	1	0.96	1	1	1	1	0.95	0.97
10M	0.86	0.98	0.98	0.99	1	0.95	1	1	1	1	0.99	0.98
2M	0.80	0.78	0.92	0.98	0.99	0.87	0.99	1	1	0.97	0.97	0.93

We also show the RB-LOH signature correlation in detail (Supplementary Figure S24). Of note, the 0.5X coverage samples raise similar signature score values to 1X and 2X. For TFs below 20%, 0.1X coverage may raise different scores and is not recommended for this approach.

Figure S24. RB-LOH signature score using different coverage in 11 samples with TF>3%.

Finally, regarding determining the sensitivity of each signature according to TF, we have:

- 1) Performed 'serial' dilutions of TF by in-silico mixing reads from the 6 cases with TF>50% with reads from the pooled 14 healthy control samples to generate TFs of: 50%, 20%, 10%, 5% and 1%. The resulting samples were analyzed with iChorCNA, and our CNA signatures were applied. We show the average correlation of the 150 signature values as a whole in each datapoint.

Table S15. Correlation of 150 CNA signatures run on samples that have been in-silico diluted to TFs of 50%, 20%, 10%, 5% and 1% (coverage 0.5X).

TF (%)	CASE 7	CASE 8	CASE 9	CASE 10	CASE 11	CASE 12	AVERAGE
50	1.00	1.00	1.00	1.00	1.00	1.00	1.00
20	0.93	0.99	0.99	0.97	0.98	0.88	0.96
10	0.86	0.96	no TF	0.92	0.92	0.48	0.83
5	0.79	0.85	0.25	0.79	no TF	0.39	0.61
1	no TF	no TF	no TF	0.39	no TF	no TF	0.39

no TF: iChorCNA reported TF value =0 and no signature values could be obtained

Results indicate that, at 1% TFs, ichorCNA may fail to detect tumor profiles in the shWGS data (5/6 fails at 1% in-silico TF). According to our data, 5-10% TF has an acceptable overall failure rate (<20%). With regards to applying our CNA signatures, we see a good correlation between samples >10% TF.

With regards to the RB-LOH signature, we observe similar results are obtained whenever ichorCNA is able to detect tumor fraction.

Figure S25. RB-LOH signature score in samples that have been in-silico diluted to TFs of 50%, 20%, 10%, 5% and 1% (coverage 0.5X).

- 54 paired samples (tissue vs plasma). Assuming that the DNA-based signatures from tissue are the gold-standard, we calculated the proportion of cases/patients where the correlation coefficient of each signature score between paired tissue versus plasma is above a certain threshold (Table S5). Overall, we observed that the vast majority of patients have a correlation coefficient >0.50. Of note, the

proportion of cases varied according to the TF, being lower in patients with TF 3-10% than TF >10%. We have now added a comment in the discussion section, limitation paragraph to reflect this limitation.

Table S5. Proportion of patients with a correlation coefficient above a certain threshold when 150 CNA-based signature scores are compared from paired tumor tissue and plasma samples.

	TF >10%	TF 3-10%	ALL
	n=29	n=25	n=54
Cor >0.80	62%	20%	43%
Cor >0.70	72%	36%	56%
Cor >0.60	86%	40%	65%
Cor >0.50	93%	40%	69%
Cor >0.40	97%	56%	78%
Cor >0.30	97%	64%	81%
Cor >0.20	97%	72%	85%
Cor >0.10	97%	76%	87%
Cor <0.10	3%	24%	13%

*, TF: tumor fraction.

3- Figure 1B does not convey any additional information. The TFs are estimated by the copy number events and therefore correlation is clearly expected. Is there any correlation between number of CNV events and an orthogonal estimate of tumor fraction? Like SNVs?

We agree with the reviewer, and we have removed Fig. 1B. The current version of Figure 1 (now Figure 2) better reflects the comparisons between tissue and plasma. Regarding correlation between number of CNV events and an orthogonal estimate of tumor fraction, we have shown in a previous comment that the correlation between ichorCNA TF estimates (which is based on number of CNV events) has a correlation >0.85 with Guardant's SNV VAF estimates (see comment above).

4- Figure 2E. This analysis is confusing. Does adjustment for TF mean a multivariable analysis with both 'RB-LOH signature' and 'TF' in the survival analysis? Also, given that there are 224 copy number segments contributing to that signature, a significance analysis is needed to show that if they randomly select 224 copy number segments, the survival stratification would no longer exist, or at least not as strong.

We apologize if this analysis is confusing. We suppose the reviewer refers to Figure 2D and not 2E, since 2D is the one showing the adjustment and the survival associations. Yes, "adjustment" means that RB-LOH signature was evaluated in a Cox model where TF was also included. We have now better clarified this.

Regarding evaluating what happens if less than 224 DNA segments are evaluated, from a prognostic perspective, we have performed the following analysis. We have randomly removed 20 DNA segments of the RB-LOH signature (measured in plasma) and evaluated its association with PFS. We have repeated this analysis 500 times and estimated the average hazard ratio and p-value. We also did the same analysis by removing 40, 80 and 160 randomly selected segments of the ctDNA-based RB-LOH signature. The results below clearly show that removing features of the RB-LOH

signature affects its prognostic ability (i.e., hazard ratio decreases and p-value increases). We have now included this result in Supplemental Material.

Supplementary Figure S26. PFS hazard ratio and p-value according to the random elimination, for 500 times, of 20, 40, 80 and 160 segments of the 224 segments of the RB-LOH signature.

5- The subtype predictor to generalize the four clusters from plasma ctDNA to tumor DNA is not properly evaluated. Authors should perform cross-validation to evaluate the performance of their proposed approach and then apply that as a classifier.

We thank the reviewer for the comment. We used consensus clustering, which is a robust approach that relies on multiple iterations of the chosen clustering method on sub-samples of the dataset. We trained the model in 178 plasma samples with TF>3% of Plasma-1 cohort using consensus clustering and validated the model in 193 plasma samples with TF>3% of Plasma-2 cohort. We also evaluate the predictor in tissue-based NGS data such as METABRIC and MSKCC cohorts, showing that the clusters 1-4 are identified and have a similar clinical behavior as in the training/plasma dataset. In the current submission, we have included the R Script to run this predictor. In the results, we state: "*Four main clusters/groups of samples were identified using consensus clustering plus (Supplementary Fig. S16) and then validated in an independent cohort of 357 plasma samples, including 193 with a TF>3% (Supplementary Fig. S17);*"

6- An analysis comparing CNV events in tumor vs plasma cfDNA is needed; perhaps similar to the one in Herberts et al (Nat 2022- Fig 3d).

We agree with the reviewer, and to further demonstrate that CNA-based data from plasma allows the identification of true biological states, we have calculated the intra-patient correlation coefficients of the CNA-based signals obtained from 514 DNA segments across 54 paired samples (tumor tissue versus plasma). Overall, 57% of patients had a correlation coefficient >0.50, which increased to 83% when 29 patients with a plasma TF >10% were evaluated. In these 29 patients with a plasma TF >10%, 59% and 24% had a correlation coefficient of >0.70 and >0.80, respectively (table below).

In addition, we have calculated the intra-patient correlation coefficients of the 150 DNA signature scores across 54 paired samples (tumor tissue versus plasma). Overall, 69% of patients had a correlation coefficient >0.50, which increased to 93% when 29 patients with a plasma TF >10% were evaluated. In these 29 patients with a plasma TF >10%, 72% and 24% had a correlation coefficient of >0.70 and >0.80, respectively (table below).

Overall, these results strongly suggest that plasma ctDNA can reliably capture CNA-based signals from tumor tissue, although the amount of ctDNA might impact the ability to accomplish this.

This limitation has now been acknowledged better in the discussion section, limitations paragraph: *"First, ~39% of patients had a TF <3%, and ctDNA-signatures could not be evaluated. In addition, ~30% of patients had a TF of 3-10% and this might limit the detection of the ctDNA-based signatures. Further studies evaluating deeper ctDNA sequencing strategies and signature detection in patients with very low TF, including those with early-stage disease, is warranted. For example, expressed genes might be inferred by evaluating nucleosome footprints from whole-genome sequencing of plasma DNA (37)."*

Table S5. Proportion of patients with a certain correlation coefficient when 514 DNA signals were compared between paired plasma and tissue.

	TF >10%	TF 3-10%	ALL
	n=29	n=25	n=54
Cor >0.80	24%	4%	15%
Cor >0.70	59%	12%	37%
Cor >0.60	62%	20%	43%
Cor >0.50	83%	28%	57%
Cor >0.40	90%	36%	65%
Cor >0.30	97%	44%	72%
Cor >0.20	97%	48%	74%
Cor >0.10	97%	64%	81%
Cor <0.10	3%	36%	19%

*, TF: tumor fraction.

Figure 2C. Correlation between plasma and tissue of 514 DNA signals in a single patient. Correlation coefficient = 0.928.

Table S5. Proportion of patients with a correlation coefficient above a certain threshold when 150 CNA-based signature scores are compared using paired tumor tissue and plasma samples.

	TF >10%	TF 3-10%	ALL
	n=29	n=25	n=54
Cor >0.80	62%	20%	43%
Cor >0.70	72%	36%	56%
Cor >0.60	86%	40%	65%
Cor >0.50	93%	40%	69%
Cor >0.40	97%	56%	78%
Cor >0.30	97%	64%	81%
Cor >0.20	97%	72%	85%
Cor >0.10	97%	76%	87%

Cor <0.10	3%	24%	13%
-----------	----	-----	-----

*, TF: tumor fraction.

8- Is the prognostic value of clusters in figure 3 more than TFs? How about total genome instability? Figure S15 shows that patients in cluster 1 have the smallest TFs, and clusters 3 & 4 have the largest. So, given the data in figure 4, this may indicate that this is about genomic instability and not necessarily the clusters found using the estimated signatures. It would be great to test this.

Yes, the prognostic value of clusters 1-4 is independent of TF in a bivariate cox model for both PFS and OS. In addition, the prognostic value of clusters 1-4 is independent of genome instability (i.e., number of CNV events). We have added both results in the legend of the current Figure 4.

9- Why are overall survivals so different between figure 3 & 4? In fact, patients in clusters 3 & 4 have very poor outcome in the first two years compared with those in the same clusters in figure 4 (HR+/HER2- tumors).

We apologize if this was not clear. Figure 3 (now Figure 4) refers to a metastatic patient population while Figure 4 (now Figure 5) refers to the METABRIC dataset, which is an early-stage patient population. Thus, the overall survival outcomes and follow-up are radically different, and this explains the differences.

10- How do expression inference methods, such as those proposed by Ulz et al (2016, 2019) or Esfahani et al (2022) work here? I would imagine one may group genes to estimate the signature activity from shWGS. I see that there is one sentence in the discussion about it, however I believe a comparison is needed to justify using CNVs and not the epigenetic footprints.

As pointed out, we make a comment in the discussion about other methods that may infer tissue gene expression; however, we believe it is beyond the scope of this study to evaluate other methods beyond CNV. Nonetheless, this is work in progress at this point, and the fact that others are trying to infer phenotypic features using blood-based methods reinforces the importance of our study and findings.

REVIEWERS' COMMENTS

Reviewer #1 (Remarks to the Author):

I have no further comments to the authors.
The comments of all three reviewers are in my mind well addressed.
The new figures well supplement the text.
My wish is that the methodology is described in the best possible way, so that the method is implemented in as many as possible laboratories as soon as possible.

Reviewer #2 (Remarks to the Author):

I commend the authors for the significant revision and thorough response to my previous comments. I have no additional comments.